# Task-Aware Virtual Training: Enhancing Generalization in Meta-Reinforcement Learning for Out-of-Distribution Tasks

Jeongmo Kim [1]  Yisak Park [1]  Minung Kim [1]  Seungyul Han [1*]

## Abstract

Meta reinforcement learning aims to develop policies that generalize to unseen tasks sampled from a task distribution. While context-based meta-RL methods improve task representation using task latents, they often struggle with out-of-distribution (OOD) tasks. To address this, we propose Task-Aware Virtual Training (TAVT), a novel algorithm that accurately captures task characteristics for both training and OOD scenarios using metric-based representation learning. Our method successfully preserves task characteristics in virtual tasks and employs a state regularization technique to mitigate overestimation errors in state-varying environments. Numerical results demonstrate that TAVT significantly enhances generalization to OOD tasks across various MuJoCo and Meta-World environments. Our code is available at https://github.com/JM-Kim-94/tavt.git.

## 1. Introduction

Research in meta reinforcement learning (meta-RL) aims to train policies on training tasks sampled from a training task distribution, with the goal of enabling the learned policy to adapt and perform well on unseen test tasks. Model-agnostic meta-learning (MAML) (Finn et al., 2017) seeks to find initial parameters that can generalize well to new tasks by using policy gradients to measure how well the current policy parameter can adapt to new tasks. On the other hand, to effectively distinguish tasks while capturing the task characteristics, context-based meta-RL methods that learn task latents through representation learning has been researched recently (Rakelly et al., 2019; Zintgraf et al., 2019; Fu et al., 2021). One of the most well-known context-based methods, PEARL (Rakelly et al., 2019), learns task representations

from off-policy samples and uses these samples for reinforcement learning, resulting in sample-efficient learning and faster convergence compared to traditional methods. In contrast, another prominent context-based meta-RL method, VariBad (Zintgraf et al., 2019), employs a Bayesian approach to learn a belief distribution over environments, effectively managing the exploration-exploitation trade-off in previously unseen environments. Recently, to address this issue and better distinguish tasks based on their characteristics, advanced representation learning methods such as contrastive learning have been increasingly adopted in meta-RL (Fu et al., 2021; Choshen & Tamar, 2023). CCM (Fu et al., 2021) is a meta-RL method that integrates contrastive learning with PEARL. In CCM, task latents from the same task are treated as having a positive relationship and are trained to be close to each other, while latents from different tasks are treated as having a negative relationship and are trained to be distant.

Advanced representation learning in meta-RL improves the ability to distinguish training tasks through learned task latents. However, most existing methods assume that the test task distribution matches the training distribution, limiting their effectiveness on out-of-distribution (OOD) tasks. Latent Dynamics Mixture (LDM) (Lee & Chung, 2021) addresses this by training policies with virtual tasks (VTs) created via linear interpolation of task latents learned by VariBad, improving OOD generalization. Despite its benefits, we identify two key issues with the existing methods for VT construction. First, generated VTs often fail to capture task characteristics accurately. Second, LDM focuses solely on reward sample generation, which struggles in environments with task-dependent state transitions. To overcome these limitations, we propose Task-Aware Virtual Training (TAVT), a novel algorithm that generates VTs accurately reflecting task characteristics for both training and OOD scenarios. Using a Bisimulation metric (Ferns et al., 2011; Ferns & Precup, 2014), our method captures task variations, such as changing goal positions, and incorporates an on-off task latent loss to stabilize the task latents. In addition, we introduce task-preserving sample generation to ensure VTs generate realistic sample contexts while maintaining task-specific features. Finally, to address state-varying environments, our task decoder generates full dynamics including

[1]Graduate School of Artificial Intelligence, UNIST, Ulsan, South Korea. * Correspondence to: Seungyul Han <syhan@unist.ac.kr>.

*Proceedings of the 42nd International Conference on Machine Learning*, Vancouver, Canada. PMLR 267, 2025. Copyright 2025 by the author(s).

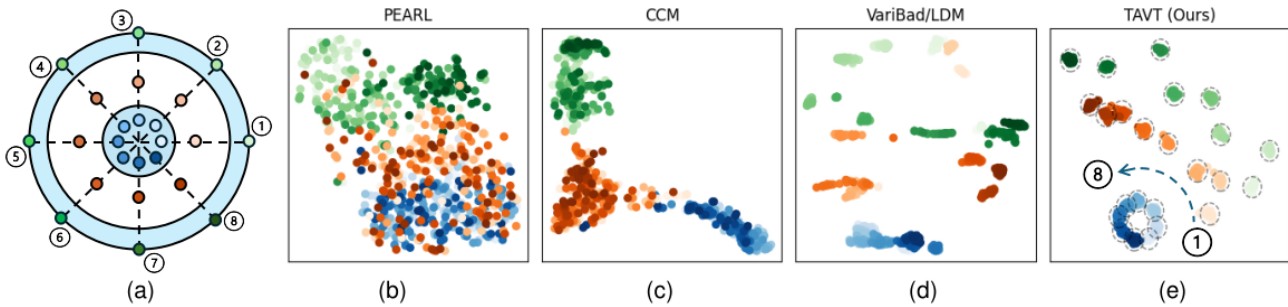

*Figure 1.* (a) 2D Goal positions in the Ant-Goal environment: The blue shaded area indicates the training task distribution, with blue marks representing inner training tasks, green marks representing outer training tasks, and red marks denoting OOD test tasks. (b-e) t-SNE visualization of task latents for various context-based meta-RL methods.

both rewards and next states, and we propose a state regularization method to mitigate overestimation errors from generated samples.

Table 1 compares our method with existing approaches, and Fig. 1 illustrates differences in task latent representations for the Ant-Goal-OOD environment, where an ant agent should reach a goal point. Fig. 1(a) shows 2D goal positions, including training and OOD test tasks. While PEARL struggles to distinguish tasks, CCM, VariBad, and LDM differentiate training tasks but scatter OOD test task representations or fail to align goal position angles. In contrast, our method (TAVT) accurately aligns task latent for both training tasks and OOD test tasks, reflecting task characteristics more effectively. In particular, while other methods either disregard VT or rely on simple reconstruction-based VT, our approach utilizes metric-based representation and generative methods, enabling the encoder to learn more accurate VT contexts that effectively capture task information. This highlights the novelty of our proposed VT framework. To introduce the proposed TAVT, the paper outlines the meta-RL setup in Section 2, our approach in Section 4, and experimental results in Section 5, showing improved task representation and OOD generalization.

*Table 1.* Comparison of Context-based Meta RL Methods

| | Task Representation | Virtual Tasks | Task-preserving VT Samples |
|---|---|---|---|
| PEARL | O | X | X |
| VariBAD | O | X | X |
| CCM | O | X | X |
| LDM | O | △ (reward only) | X |
| TAVT (Ours) | O | O | O |

## 2. Preliminary

### 2.1. Meta Reinforcement Learning

In meta-RL, each task $\mathcal{T}$ is sampled from a task distribution $p(\mathcal{T})$ and defined as a Markov Decision Process (MDP) $(S, A, P^{\mathcal{T}}, R^{\mathcal{T}}, \gamma, \rho_0)$, where $S$ and $A$ are the state and action spaces, $P^{\mathcal{T}}$ represents state transition dynamics, $R^{\mathcal{T}}$ is the reward function, $\gamma \in [0, 1)$ is the discount factor, and

$\rho_0$ is the initial state distribution. At each time step $t$, the agent selects an action $a_t$ based on the policy $\pi$, receives a reward $r_t := R^{\mathcal{T}}(s_t, a_t)$, and transitions to the next state $s_{t+1} \sim P^{\mathcal{T}}(\cdot|s_t, a_t)$. The MDP for each task may vary, but all tasks share the same state and action spaces. During meta-training, the policy $\pi$ is optimized to maximize the cumulative reward sum $\sum_t \gamma^t r_t$ across tasks sampled from $p(\mathcal{T}_{\text{train}})$. The policy is then evaluated on OOD test tasks from $p(\mathcal{T}_{\text{test}})$, which differs entirely from $p(\mathcal{T}_{\text{train}})$.

### 2.2. Context-based Meta RL

Recent meta-RL methods focus on learning latent contexts to differentiate tasks. PEARL (Rakelly et al., 2019), a well-known context-based meta-RL approach, learns task latents $\mathbf{z} \sim q_\psi(\cdot|\mathbf{c}^{\mathcal{T}})$ using a task encoder $q_\psi$ with parameters $\psi$ and task context $\mathbf{c}^{\mathcal{T}} := \{(s_l, a_l, r_l, s'_l)\}_{l=1}^{N_c}$, where $s'$ is the next state and $N_c$ is the number of transition samples. PEARL defines task-dependent policies $\pi(\cdot|s, \mathbf{z})$ and Q-functions $Q(s, a, \mathbf{z})$, using soft actor-critic (SAC) (Haarnoja et al., 2018) to train policies. The task encoder $q_\psi$ is trained to minimize the encoder loss:

$$\mathbb{E}_{\mathcal{T} \sim p(\mathcal{T}_{\text{train}})}[\mathbb{E}_{\mathbf{z} \sim q_\psi}[\mathcal{L}_Q(\mathcal{T}, \mathbf{z})] + D_{\text{KL}}(q_\psi(\cdot|\mathbf{c}^{\mathcal{T}})||N(\mathbf{0}, \mathbf{I}))],$$

where $N$ is the multivariate Gaussian distribution, $\mathbf{I}$ is the identity matrix, and $\mathcal{L}_Q$ is the SAC critic loss. Inspired by the variational auto-encoder (VAE) (Kingma & Welling, 2013), PEARL uses $\mathcal{L}_Q$ instead of VAE's reconstruction loss to better distinguish tasks.

### 2.3. Virtual Task Construction

To improve the generalization of policies to various OOD tasks, LDM (Lee & Chung, 2021) defines the task latents $\mathbf{z}^\alpha$ for a virtual task as a linear interpolation of training task latents $\mathbf{z}^i$, $i = 1, \cdots, M$, expressed as:

$$\mathbf{z}^\alpha = \sum_{i=1}^{M} \alpha^i \mathbf{z}^i \tag{1}$$

where $\alpha = (\alpha^1, \cdots, \alpha^M) \sim \beta\text{Dirichlet}(1, 1, ..., 1) - \frac{\beta-1}{M}$ is the interpolation coefficient, $\text{Dirichlet}(\cdot)$ is the Dirichlet distribution, and $M$ is the number of training tasks used for mixing. The parameter $\beta \geq 1$ controls the degree of mixing: with $\beta = 1$, only interpolation within the training task latents occurs, while $\beta > 1$ allows extrapolation beyond the original latents. LDM further trains the policy using contexts generated from the task decoder based on the interpolated latents $\mathbf{z}^\alpha$, enabling it to handle OOD tasks more effectively.

## 3. Related Works

**Advanced Task Representation Learning:** Advanced representation learning techniques have been widely explored to improve task latents that effectively distinguish tasks. Recent meta-RL methods use contrastive learning (Oord et al., 2018) to enhance task differentiation through positive and negative pairs, improving task representation (Laskin et al., 2020; Fu et al., 2021; Choshen & Tamar, 2023) and capturing task information in offline setups (Li et al., 2020b; Gao et al., 2023). The Bisimulation metric (Ferns et al., 2011) is employed to capture behavioral similarities (Zhang et al., 2021; Agarwal et al., 2021; Liu et al., 2023) and group similar tasks (Hansen-Estruch et al., 2022; Sodhani et al., 2022). Additionally, skill representation learning (Eysenbach et al., 2018) addresses non-parametric meta-RL challenges (Frans et al., 2017; Harrison et al., 2020; Nam et al., 2022; Fu et al., 2022; He et al., 2024), while task representation learning is increasingly applied in multi-task setups (Ishfaq et al., 2024; Cheng et al., 2022; Sodhani et al., 2021).

**Generalization for OOD Tasks:** Meta-RL techniques for improving policy generalization in OOD test environments have been actively studied (Lan et al., 2019; Fakoor et al., 2019; Mu et al., 2022). Model-based approaches (Lin et al., 2020; Lee & Chung, 2021), advanced representation learning with Gaussian Mixture Models (Wang et al., 2023b; Lee et al., 2023), and Transformers (Vaswani et al., 2017; Melo, 2022; Xu et al., 2024) have been explored. Additionally, some studies tackle distributional shift challenges through robust learning (Mendonca et al., 2020; Mehta et al., 2020; Ajay et al., 2022; Greenberg et al., 2023).

**Model-based Sample Relabeling:** Model-based sample generation and relabeling techniques have gained attention in meta-RL (Rimon et al., 2024; Wen et al., 2024), enabling the reuse of samples from other tasks using dynamics models (Li et al., 2020a; Mendonca et al., 2020; Wan et al., 2021; Zou et al., 2024). These methods address sparse rewards (Packer et al., 2021; Jiang et al., 2023), mitigate distributional shifts in offline setups (Dorfman et al., 2021; Yuan & Lu, 2022; Zhou et al., 2024; Guan et al., 2024), and incorporate human preferences (Ren et al., 2022; Hejna III & Sadigh, 2023) or guided trajectory relabeling (Wang et al., 2023a), expanding their applications.

## 4. Methodology

### 4.1. Metric-based Task Representation

In this section, we propose a novel representation learning method to ensure task latents accurately capture differences in task contexts, enabling virtual tasks to effectively reflect task characteristics. To achieve this, we leverage the Bisimulation metric (Ferns et al., 2011), which measures the similarity of two states in an MDP based on the reward function $R^\mathcal{T}$ and state transition $P^\mathcal{T}$. In meta-learning, the Bisimulation metric can quantify task similarity by comparing contexts (Zhang et al., 2021). Unlike Zhang et al. (2021), which considers tasks with different state spaces, we adapt the metric for tasks sharing the same state and action space, modifying it from Eq. (4) in Zhang et al. (2021).

**Definition 4.1** (Bisimulation metric for task representation).
For two different tasks $\mathcal{T}_i$ and $\mathcal{T}_j$,

$$d(\mathcal{T}_i, \mathcal{T}_j) = \underset{(s,a)\sim D}{\mathbb{E}}\Big[|R^{\mathcal{T}_i}(s,a) - R^{\mathcal{T}_j}(s,a)| + \eta W_2(P^{\mathcal{T}_i}(\cdot|s,a), P^{\mathcal{T}_j}(\cdot|s,a))\Big], \quad (2)$$

where $D$ is the replay buffer that stores the sample contexts, $R^\mathcal{T}, P^\mathcal{T}$ are the reward function and the transition dynamics for task $\mathcal{T}$, $W_2$ is 2-Wasserstein distance between the two distributions, and $\eta \in (0, 1]$ is the distance coefficient.

**Proposition 4.2.** $d(\cdot, \cdot)$ *defined in Eq.* (2) *is a metric.*

Proof) The detailed proof is provided in Appendix A.

The Bisimulation metric $d$ equals $0$ when the contexts of two tasks perfectly match and increases as the context difference grows. Task latents learned using this metric more effectively capture task distances than existing representation methods, as shown in Fig. 1. We train the task encoder $q_\psi(\mathbf{z}|\mathbf{c}^\mathcal{T})$ to ensure the task latent $\mathbf{z} \sim q_\psi$ preserves the Bisimulation metric $d$ in the latent space. Since the actual reward function $R$ and transition dynamics $P$ are generally unknown, we train the task decoder $p_\phi(s, a, \mathbf{z}) = \big(R_\phi(s, a, \mathbf{z}), P_\phi(\cdot|s, a, \mathbf{z})\big)$ with parameters $\phi$ to approximate task dynamics using reconstruction loss.

We adopt the learning structure of PEARL, which uses two distinct policies: $\pi_{\exp}$ for on-policy exploration to obtain task latents from contexts and $\pi_{\mathrm{RL}}$ for off-policy RL to maximize returns. Contexts generated by $\pi_{\exp}$ and $\pi_{\mathrm{RL}}$ are stored in the on-policy buffer $D_{\mathrm{on}}^\mathcal{T}$ and the off-policy buffer $D_{\mathrm{off}}^\mathcal{T}$, respectively. PEARL considers only on-policy task latents $\mathbf{z}_{\mathrm{on}}$ derived from $\mathbf{c}^\mathcal{T} \sim D_{\mathrm{on}}^\mathcal{T}$, but $\mathbf{z}_{\mathrm{on}}$ can be unstable due to limited contexts in $D_{\mathrm{on}}^\mathcal{T}$. To address this, we propose an on-off latent learning structure where off-policy task latents $\mathbf{z}_{\mathrm{off}}$, derived from $\mathbf{c}_{\mathrm{off}}^\mathcal{T} \sim D_{\mathrm{off}}^\mathcal{T}$, maintain the Bisimulation distance for training tasks, and $\mathbf{z}_{\mathrm{on}}$ just aligns with $\mathbf{z}_{\mathrm{off}}$. Fig. 2 shows t-SNE visualization of $\mathbf{z}_{\mathrm{on}}$, demonstrating that on-off latent loss provides more stable task

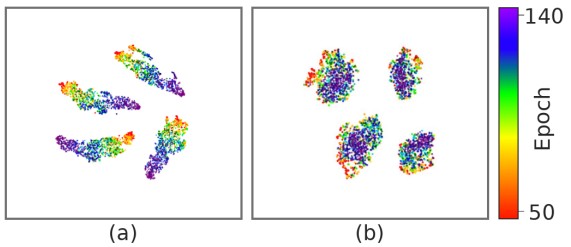

(a)    (b)

*Figure 2.* Changes in latents $\mathbf{z}_{\text{on}}$ of 4 randomly sampled training tasks in the Ant-Goal-OOD environment: (a) Using $\mathbf{z}_{\text{on}}$ only (b) Using on-off latent loss for task representation learning

representation compared to using $\mathbf{z}_{\text{on}}$ alone. In summary, our encoder-decoder loss is defined as

$$\mathcal{L}_{\text{bisim}}(\psi, \phi) = \mathbb{E}_{\mathcal{T}_i, \mathcal{T}_j \sim p(\mathcal{T}_{\text{train}})} \Big[ \underbrace{\Big( |\mathbf{z}_{\text{off}}^i - \mathbf{z}_{\text{off}}^j| - d(\mathcal{T}_i, \mathcal{T}_j; p_{\bar{\phi}}) \Big)^2}_{\text{Bisimulation loss}}$$

$$+ \underbrace{\mathbb{E}_{(s,a,r,s') \sim D_{\text{off}}^{\mathcal{T}_i}, (\hat{r}, \hat{s}') \sim p_\phi(s, a, \mathbf{z}_{\text{off}}^i)} \Big[ (r - \hat{r})^2 + (s' - \hat{s}')^2 \Big]}_{\text{Reconstruction loss}}$$

$$+ \underbrace{(\mathbf{z}_{\text{on}}^i - \bar{\mathbf{z}}_{\text{off}}^i)^2}_{\text{on-off latent loss}} \Big], \quad \mathbf{z}^i \sim q_\psi(\cdot | \mathbf{c}^{\mathcal{T}_i}), \ \mathbf{c}^{\mathcal{T}_i} \sim D^{\mathcal{T}_i}, \ \forall i, \quad (3)$$

where $d(\mathcal{T}_i, \mathcal{T}_j; p_{\bar{\phi}})$ replaces $(R^{\mathcal{T}_i}, P^{\mathcal{T}_i})$ in Eq. (2) with the task decoder $p_{\bar{\phi}}(\cdot, \bar{\mathbf{z}}_{\text{off}}^i), \forall i$, and $\bar{x}$ denotes gradient detachment for $x$. We construct the task latents $\mathbf{z}^\alpha$ for VTs using the method in Section 2. Our proposed representation learning ensures task latents align more effectively to capture task differences based on the Bisimulation distance $d$, as illustrated in Fig.1. Additionally, in Section 5.3 and Appendix B, we present task representations for other tasks, demonstrating that our method consistently aligns and stabilizes task latents across various environments.

### 4.2. Task Preserving Sample Generation

Using the task decoder $p_\phi(\cdot, \mathbf{z}^\alpha)$ and VT task latents $\mathbf{z}^\alpha$, we generate virtual contexts $\hat{\mathbf{c}}^\alpha := (s_l, a_l, \hat{r}_l^\alpha, \hat{s}_l'^\alpha)_{l=1}^{N_c}$, where $(s_l, a_l)$ are sampled from real contexts $\mathbf{c}^{\mathcal{T}}$, and $\hat{r}_l^\alpha, \hat{s}_l'^\alpha \sim p_\phi(s_l, a_l, \mathbf{z}^\alpha)$. Existing VT construction methods (Lee & Chung, 2021; Lee et al., 2023) use dropout-based regularization to ensure virtual contexts generalize to unseen tasks. However, we observed that task latents $\hat{\mathbf{z}}^\alpha \sim q_\psi(\cdot | \hat{\mathbf{c}}^\alpha)$ often deviate significantly from $\mathbf{z}^\alpha$, indicating that virtual contexts fail to effectively preserve task information. To address this, we propose a task-preserving loss to minimize the difference between $\mathbf{z}^\alpha$ and $\hat{\mathbf{z}}^\alpha$, ensuring virtual contexts better retain task latent information. Fig. 3 demonstrates the effectiveness of the task-preserving loss. In Fig. 3(a), without the task-preserving loss, $\mathbf{z}^\alpha$ and $\hat{\mathbf{z}}^\alpha$ show significant differences, resulting inaccurate task latents for OOD tasks. In contrast, Fig. 3(b) shows that with the task-preserving loss, $\mathbf{z}^\alpha$ and $\hat{\mathbf{z}}^\alpha$ align closely, enhancing stability and alignment for OOD tasks.

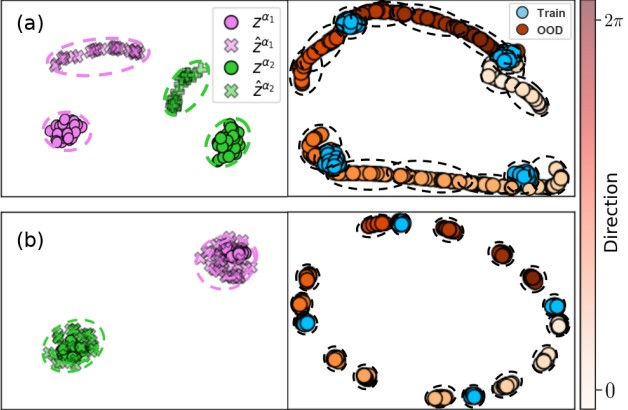

*Figure 3.* Comparison of task latents in the Ant-Dir-4 Environment: (a) Without task-preserving loss (b) With task-preserving loss. The left graph shows the difference between latents $\mathbf{z}^\alpha$ and $\hat{\mathbf{z}}^\alpha$ for VTs during training, while the right graph shows latents for training (blue circles) and OOD tasks (red circles) during evaluation. Training tasks have 4 goal directions of $\frac{\pi}{4}k$, $k = 0, 1, 2, 3$, and OOD tasks cover 12 different directions from 0 to $2\pi$.

Despite the benefits of the task-preserving loss, virtual contexts $\hat{\mathbf{c}}^\alpha$ may still differ from real contexts due to limitations in the task decoder $p_\phi(\cdot, \mathbf{z}^\alpha)$, which cannot fully capture actual task contexts. These differences can introduce instability and degrade performance for RL training. To address this issue, we use a Wasserstein generative adversarial network (WGAN) (Arjovsky et al., 2017), designed to reduce the distribution gap between real and generated data. In our setup, the task decoder $p_\phi(\cdot, \mathbf{z}^\alpha)$ acts as the generator, learning to produce samples that closely resemble real ones, while real contexts serve as the target for the discriminator. The discriminator $f_\zeta$ increases its value for real samples and decreases it for generated samples, while the generator aligns virtual contexts with real ones by increasing $f_\zeta$ for generated samples. This process ensures that virtual contexts not only preserve task information but also closely resemble real contexts, significantly reducing the gap between VT-generated samples and real OOD task samples. To summarize, the WGAN discriminator and generator losses, combined with the task-preserving loss, are defined as:

$$\mathcal{L}_{\text{disc}}(\zeta) = \mathbb{E}_{\mathcal{T}_i \sim p(\mathcal{T}_{\text{train}}), \mathbf{c}^{\mathcal{T}_i} \sim D^{\mathcal{T}_i}} [-f_\zeta(\mathbf{c}^{\mathcal{T}_i}, \bar{\mathbf{z}}_{\text{off}}^i) \quad (4)$$

$$+ \mathbb{E}_{\hat{\mathbf{c}}^\alpha \sim p_\phi} [f_\zeta(\hat{\mathbf{c}}^\alpha, \bar{\mathbf{z}}_{\text{off}}^\alpha)]] + \lambda_{\text{GP}} \cdot \text{Gradient Penalty},$$

$$\mathcal{L}_{\text{gen}}(\psi, \phi) = \underbrace{\mathbb{E}_{\hat{\mathbf{c}}^\alpha \sim p_\phi} [-f_\zeta(\hat{\mathbf{c}}^\alpha, \bar{\mathbf{z}}_{\text{off}}^\alpha)}_{\text{WGAN generator loss}}$$

$$+ \underbrace{\mathbb{E}_{\hat{\mathbf{z}}^\alpha \sim q_\psi(\cdot | \hat{\mathbf{c}}^\alpha)} [(\hat{\mathbf{z}}^\alpha - \bar{\mathbf{z}}_{\text{off}}^\alpha)^2]]}_{\text{task preserving loss}}, \quad (5)$$

where the Gradient Penalty (GP), introduced in (Arjovsky et al., 2017), stabilizes training, with $\lambda_{\text{GP}}$ as its coefficient. The detailed implementation of GP is provided in Appendix D.2. In this framework, the task decoder $p_\phi$ is trained with

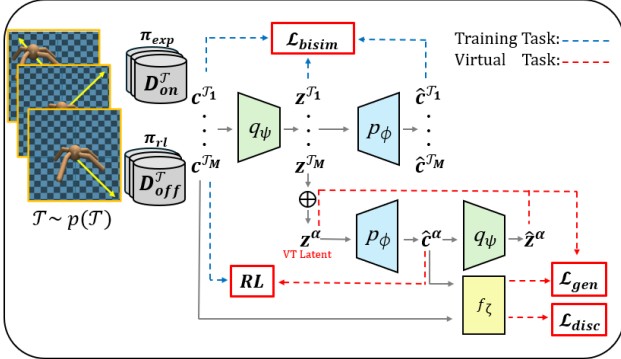

*Figure 4.* An illustration for the structure of TAVT

input $z_{off}$, as described in Eq. (3). Both the task-preserving loss and VT construction are always based on $z_{off}^{\alpha}$, derived from $z_{off}$. Importantly, the off-policy task latent $z_{off}$ conditions sample context generation, with its gradient disconnected to ensure stable training. By leveraging WGAN, we significantly reduce differences between generated and real samples, improving RL performance on OOD tasks. In Section 5.4 and Appendix C, we demonstrate that the proposed task-preserving sample generation effectively bridges the gap between VT-generated samples and real OOD task samples, enhancing generalization and RL performance.

### 4.3. Task-Aware Virtual Training

By combining metric-based representation learning with task-preserving sample generation, we propose the Task-Aware Virtual Training (TAVT) algorithm, which enhances the generalization of meta-RL to OOD tasks by leveraging VTs that accurately reflect task characteristics. The total encoder-decoder loss for TAVT is given by:

$$\mathcal{L}_{total}(\psi, \phi) = \mathcal{L}_{bisim}(\psi, \phi) + \mathcal{L}_{gen}(\psi, \phi), \qquad (6)$$

with detailed loss scales provided in Appendix F.

Now, we train the RL policy $\pi_{RL}$ using SAC, leveraging both real contexts $c^{\mathcal{T}_i} \sim D_{off}^{\mathcal{T}_i}$ for each training task $\mathcal{T}_i$ and virtual contexts $\hat{c}^{\alpha}$ generated by the proposed TAVT method. For training tasks, the RL policy is defined as $\pi_{RL} = \pi(\cdot|s, z_{on}^i)$, where $z_{on}^i$ is the on-policy task latent. For VTs, the RL policy is defined as $\pi_{RL} = \pi(\cdot|s, z_{on}^{\alpha})$, where $z_{on}^{\alpha}$ is the VT's task latent derived from $z_{on}^i$. We train the RL policy $\pi_{RL}$ using SAC, leveraging both real contexts $c^{\mathcal{T}_i} \sim D_{off}^{\mathcal{T}_i}$ for each training task $\mathcal{T}_i$ and virtual contexts $\hat{c}^{\alpha}$ generated by the proposed TAVT method. For training tasks, the RL policy is defined as $\pi_{RL} = \pi(\cdot|s, z_{on}^i)$, where $z_{on}^i$ is the on-policy task latent. For VTs, the RL policy is defined as $\pi_{RL} = \pi(\cdot|s, z_{on}^{\alpha})$, where $z_{on}^{\alpha}$ is the VT's task latent derived from $z_{on}^i$. The proposed TAVT method aligns task latents using metric-based representation and learns VTs that preserve task characteristics while generating sam-

ples similar to real ones through task-preserving loss with WGAN. This enables the policy to train on a diverse range of OOD tasks, improving generalization performance.

Since the agent cannot access off-policy latents for test tasks, it relies on on-policy latents $z_{on}^i$, obtained from $N_{exp}$ episodes generated by the exploration policy $\pi_{exp}$, as introduced in PEARL (Rakelly et al., 2019). While PEARL uses $\pi(\cdot|s, \tilde{z})$ with $\tilde{z} \sim N(\mathbf{0}, \mathbf{I})$ as the exploration policy, we propose a novel exploration policy $\pi_{exp} = \pi(\cdot|s, z_{on}^{\alpha})$ that leverages VT task latents $z_{on}^{\alpha}$. This approach enables exploration across both training tasks and VTs by varying the interpolation coefficient $\alpha$, allowing the agent to explore a broader range of tasks. Appendix E shows that our exploration policy covers a wider range of trajectories, including OOD tasks, leading to improved generalization. Fig. 4 provides an overview of the TAVT framework, with detailed implementation, including the RL loss function and meta-training/testing algorithms, in Appendix D.3.

### 4.4. State Regularization Method

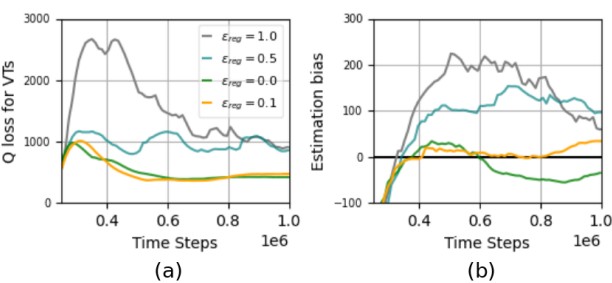

*Figure 5.* (a) $Q$-function loss for VTs (b) Estimation bias for OOD tasks in the Walker-Mass-OOD environment.

The virtual contexts $\hat{c}^{\alpha}$ in TAVT include both rewards and next states, enabling it to handle state-varying environments. However, inaccuracies in the task decoder can introduce errors in the $Q$-function, leading to overestimation bias, a common issue in offline RL (Fujimoto et al., 2019). While reward errors have minimal impact, next-state errors significantly contribute to overestimation. To mitigate this, we propose a state regularization method, replacing the next states $\hat{s}'^{\alpha}$ in virtual contexts with $\hat{s}_{reg}'^{\alpha}$, a mix of next states $s'$ from training tasks and $\hat{s}'^{\alpha}$ from virtual contexts:

$$\hat{s}_{reg}'^{\alpha} := \epsilon_{reg}\hat{s}'^{\alpha} + (1 - \epsilon_{reg})s',$$

where $(s, a, r, s') \in c^{\mathcal{T}}$, $\hat{r}^{\alpha}, \hat{s}'^{\alpha} \sim p_{\phi}(s, a, z_{off}^{\alpha})$, and $\epsilon_{reg} \in [0, 1]$ is the regularization coefficient.

Fig. 5(a) shows the $Q$-function loss for VTs during training, while Fig. 5(b) illustrates the $Q$-function estimation bias for OOD tasks in the Walker-Mass-OOD environment, where the agent's mass varies by task. Estimation bias, defined as the difference between $Q$-function values and average actual returns, is significantly reduced with $\epsilon_{reg} = 0.1$.

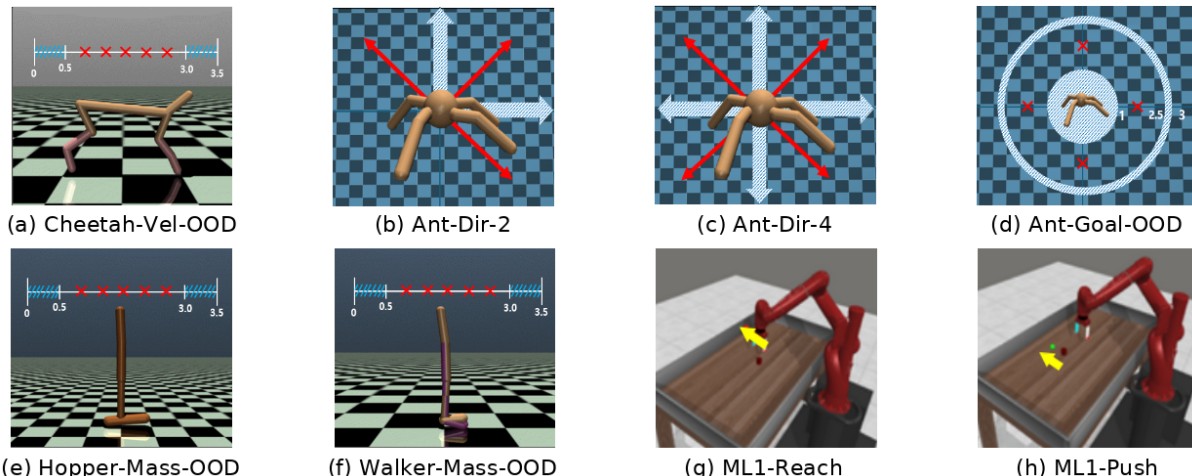

(a) Cheetah-Vel-OOD      (b) Ant-Dir-2      (c) Ant-Dir-4      (d) Ant-Goal-OOD

(e) Hopper-Mass-OOD      (f) Walker-Mass-OOD      (g) ML1-Reach      (h) ML1-Push

*Figure 6.* (a-f) MuJoCo environments (g-h) ML1 environments

In contrast, using $\epsilon_{\text{reg}} = 1.0$ (full use of $\hat{s}'^{\alpha}$) results in higher $Q$-function loss and greater bias. Section 5 further demonstrates that this method improves OOD performance.

## 5. Experiments

In this section, we compare the proposed TAVT algorithm with various on-policy and off-policy meta-RL methods across MuJoCo (Todorov et al., 2012) and MetaWorld ML1 (Yu et al., 2020) environments. For off-policy methods, we include PEARL (Rakelly et al., 2019), CCM (Fu et al., 2021) which uses contrastive learning to enhance representation, MIER (Mendonca et al., 2020) which incorporates a gradient-based dynamics model, and Amago (Grigsby et al., 2024) which employs transformers for latent learning. For on-policy methods, we evaluate MAML (Finn et al., 2017) which optimizes initial gradients for generalization, RL$^2$ (Duan et al., 2016) which encodes task information in RNN hidden states, VariBad (Zintgraf et al., 2019) which applies Bayesian methods for representation learning, and LDM (Lee & Chung, 2021) which uses virtual tasks with a reconstruction loss and a DropOut layer. Additionally, we provide an analysis of the task representation in TAVT as well as an ablation study on its performance. For TAVT, detailed hyperparameter configurations are provided in Appendix F. For other methods, we reproduced their results using the author-provided source code and default hyperparameters. Comparison graphs and tables show the average performance over 5 random seeds, with standard deviations as shaded areas in the graphs or $\pm$ in the tables.

### 5.1. Environmental Setup

To evaluate generalization performance on OOD tasks, we used 6 MuJoCo environments (Todorov et al., 2012): Cheetah-Vel-OOD, Ant-Dir-2, Ant-Dir-4, and Ant-Goal-

*Table 2.* Environmental setup

| | $\mathcal{M}_{\text{train}}$ | $\mathcal{M}_{\text{test}}$ |
|---|---|---|
| Cheetah-Vel-OOD | $[0.0, 0.5) \cup [3.0, 3.5)$ | $\{0.75, 1.25, 1.75, 2.25, 2.75\}$ |
| Ant-Dir-2 | $\{0, \frac{\pi}{2}\}$ | $\{\frac{\pi}{4}, \frac{3\pi}{4}, \frac{7\pi}{4}\}$ |
| Ant-Dir-4 | $\{0, \frac{\pi}{2}, \pi, \frac{3\pi}{2}\}$ | $\{\frac{\pi}{4}, \frac{3\pi}{4}, \frac{5\pi}{4}, \frac{7\pi}{4}\}$ |
| Ant-Goal-OOD | $r \in [0.0, 1.0) \cup [2.5, 3.0)$ | $r = 1.75$ |
| | $\theta \in [0, 2\pi]$ | $\theta \in \{0, \frac{\pi}{2}, \pi, \frac{3\pi}{2}\}$ |
| Hopper-Mass-OOD | $[0.0, 0.5) \cup [3.0, 3.5)$ | $\{0.75, 1.25, 1.75, 2.25, 2.75\}$ |
| Walker-Mass-OOD | $[0.0, 0.5) \cup [3.0, 3.5)$ | $\{0.75, 1.25, 1.75, 2.25, 2.75\}$ |
| ML1 | $\mathcal{M}$ | $\mathcal{M}$ |
| ML1-OOD-Inter | $\mathcal{M} \backslash \mathcal{M}_{\text{inner}}$ | $\mathcal{M}_{\text{inner}}$ |
| ML1-OOD-Extra | $\mathcal{M}_{\text{inner}}$ | $\mathcal{M} \backslash \mathcal{M}_{\text{inner}}$ |

OOD, where only the reward function varies across tasks, and Walker-Mass-OOD and Hopper-Mass-OOD, where both the reward function and state transition dynamics vary. In addition, we considered 6 MetaWorld ML1 environments (Yu et al., 2020), including the original Reach and Push environments, as well as 4 OOD variations (Reach-OOD-Inter, Reach-OOD-Extra, Push-OOD-Inter, and Push-OOD-Extra). In these ML1 and ML1-OOD tasks, the final goal point varies across tasks while maintaining shared state dynamics. The task space is denoted by $\mathcal{M}$, and detailed descriptions of these environments are provided below.

- Cheetah-Vel-OOD: The Cheetah agent is required to run at target velocities $v_{\text{tar}}$ in $\mathcal{M}$.

- Ant-Dir: The Ant agent is required to move in directions $\theta_{\text{dir}}$ in $\mathcal{M}$. For Ant-Dir-2, the training tasks involve 2 directions ($\theta_{\text{dir}} = 0, \frac{\pi}{2}$), and for Ant-Dir-4, the training tasks involve 4 directions ($\theta_{\text{dir}} = 0, \frac{\pi}{2}, \pi, \frac{3\pi}{2}$).

- Ant-Goal-OOD: The Ant agent should reach the 2D goal positions $(r_{\text{goal}} \cos \theta_{\text{goal}}, r_{\text{goal}} \sin \theta_{\text{goal}})$, where the radius $r_{\text{goal}}$ and angle $\theta_{\text{goal}}$ are selected from $\mathcal{M}$.

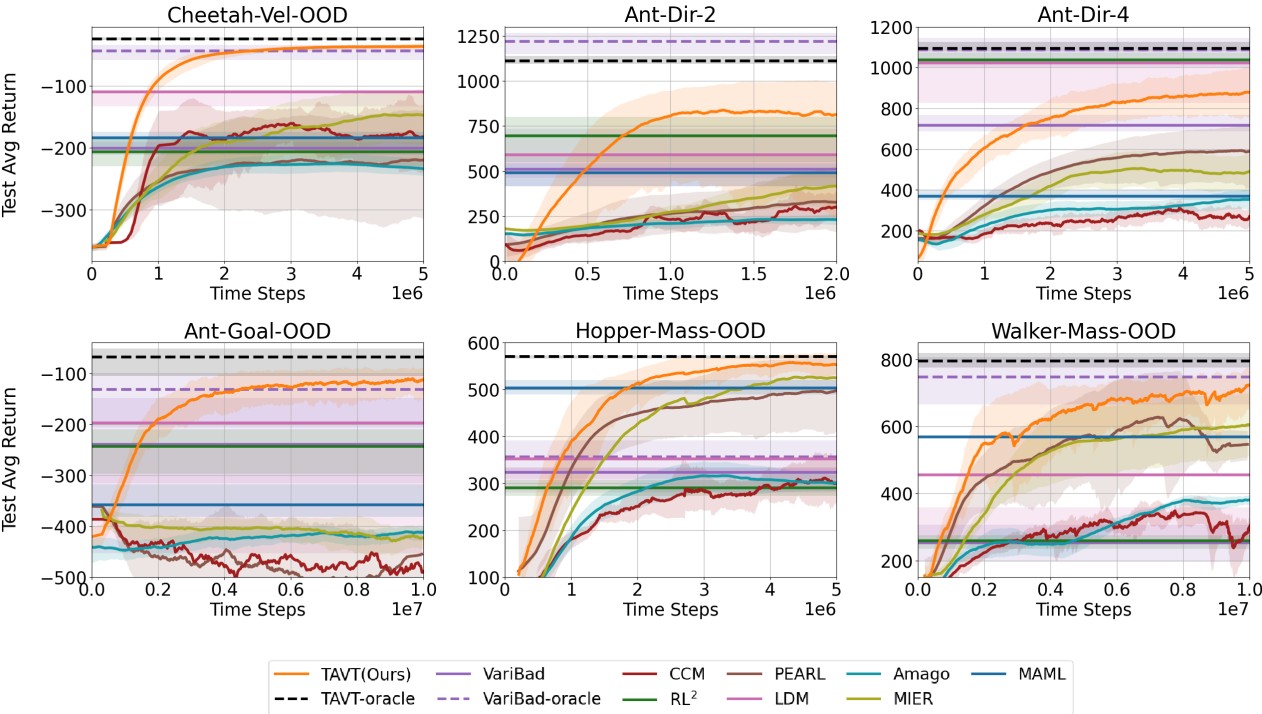

*Figure 7.* Performance comparison for MuJoCo environments. The graphs for on-policy algorithms represent their final performance.

- Hopper/Walker-Mass-OOD: The Hopper/Walker agent are required to run forward with scale $m_{\text{scale}}$ in $\mathcal{M}$ multiplied to their body mass.
- ML1/ML1-OOD: The agent is tasked with reaching or pushing an object to a target goal position $g_{\text{tar}}$, sampled from the 3D goal space $\mathcal{M}$, which varies depending on the environment. To create an OOD setup, inner areas $\mathcal{M}_{\text{inner}}$ are defined within the goal space $\mathcal{M}$.

The training task space $\mathcal{M}_{\text{train}}$ and test task space $\mathcal{M}_{\text{test}}$ are detailed in Table 2. Except for the original ML1 environments, test tasks are OOD, meaning $\mathcal{M}_{\text{train}} \bigcap \mathcal{M}_{\text{test}} = \emptyset$. For Ant-Dir-2, test tasks include both interpolations between training directions and extrapolations beyond them. In ML1-OOD-Extra, OOD test tasks ($\mathcal{M} \backslash \mathcal{M}_{\text{inner}}$) extrapolate beyond training tasks ($\mathcal{M}_{\text{inner}}$). In all other OOD tasks, test tasks lie in regions interpolating between training tasks. During training, tasks are uniformly sampled from $\mathcal{M}_{\text{train}}$, with $p(\mathcal{T}_{\text{train}}) = \text{Unif}(\mathcal{M}_{\text{train}})$. A visualization of the environments is provided in Fig. 6. Further details on MuJoCo and MetaWorld, including the definitions of the goal spaces $\mathcal{M}$ and $\mathcal{M}_{\text{inner}}$ for ML1/ML1-OOD, are available in Appendix G.

**5.2. Performance Comparison**

We compare the performance of the proposed TAVT with other meta-RL algorithms. For MuJoCo environments, Fig.

7 shows the convergence performance of the test average return, while Table 1 presents the final average success rate for ML1/ML1-OOD environments. On-policy algorithms are trained for 200M timesteps in MuJoCo and 100M timesteps in MetaWorld environments. For off-policy algorithms, training timesteps vary in MuJoCo environments and are fixed at 10M timesteps for MetaWorld environments. Also, Fig. 7 includes the '-oracle' performance for VariBad (on-policy) and TAVT (off-policy) in MuJoCo environments. This represents the upper performance bound achieved by training on all tasks from both $\mathcal{M}_{\text{train}}$ and $\mathcal{M}_{\text{test}}$.

For MuJoCo environments, Fig. 7 shows that both VariBad and TAVT perform well in the '-oracle' setup across most environments. However, in OOD task scenarios, TAVT significantly outperforms other on-policy and off-policy methods. Notably, TAVT achieves performance close to the '-oracle' in Cheetah-Vel-OOD, Ant-Goal-OOD, and Walker-Mass-OOD, demonstrating strong generalization to unseen OOD tasks. While other algorithms struggle in challenging environments like Ant-Dir-2, which includes extrapolation tasks, TAVT remains robust. In environments such as Walker-Mass-OOD and Hopper-Mass-OOD, where state transition dynamics vary, LDM fails to adapt due to limitations in its VT construction. In contrast, TAVT excels, showcasing its ability to handle varying state transition dynamics. Details on policy trajectories for training and OOD tasks are available in Appendix E.

*Table 3.* Average success rate for MetaWorld ML1 environments.

|  | MAML | RL[2] | VariBAD | LDM | PEARL | CCM | Amago | MIER | TAVT(Ours) |
|---|---|---|---|---|---|---|---|---|---|
| Reach | 0.97±0.02 | 0.95±0.04 | 0.73±0.12 | 0.76±0.1 | 0.48±0.21 | 0.65±0.13 | 0.71±0.27 | 0.61±0.18 | **0.98±0.02** |
| Reach-OOD-Inter | 0.56±0.11 | 0.86±0.12 | 0.82±0.11 | 0.87±0.1 | 0.52±0.16 | 0.78±0.1 | 0.93±0.05 | 0.62±0.18 | **0.96±0.03** |
| Reach-OOD-Extra | 0.48±0.15 | 0.73±0.14 | 0.82±0.11 | 0.79±0.15 | 0.48±0.14 | 0.81±0.12 | 0.43±0.08 | 0.65±0.12 | **0.99±0.01** |
| Push | 0.94±0.03 | **0.98±0.02** | 0.88±0.09 | 0.83±0.11 | 0.61±0.11 | 0.18±0.08 | 0.87±0.11 | 0.59±0.13 | **0.98±0.03** |
| Push-OOD-Inter | 0.78±0.13 | 0.79±0.14 | 0.83±0.11 | 0.77±0.13 | 0.79±0.16 | 0.12±0.03 | **0.98±0.02** | 0.45±0.15 | **0.98±0.02** |
| Push-OOD-Extra | 0.55±0.13 | 0.38±0.12 | 0.65±0.09 | 0.72±0.11 | 0.55±0.18 | 0.15±0.04 | 0.83±0.11 | 0.61±0.15 | **0.92±0.08** |

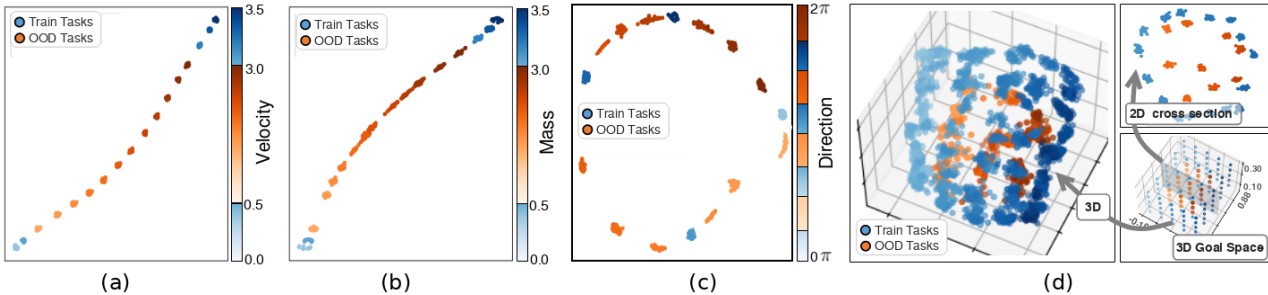

*Figure 8.* t-SNE visualization of task latents: (a) Cheetah-Vel-OOD (b) Walker-Mass-OOD (c) Ant-Dir-4 (d) Reach-OOD-Inter. For ML1 tasks in the 3D goal space, we provide both 3D representation for all tasks and 2D representation for tasks in the selected cross-section.

For MetaWorld environments, Table 3 shows that TAVT consistently delivers superior performance across all setups. In the original ML1 environments, TAVT outperforms other methods, highlighting the advantages of its metric-based task representation. In ML1-OOD environments, PEARL-based methods such as PEARL, CCM, and MIER perform poorly in Push and Push-OOD tasks, while other off-policy algorithms also fail to achieve success rates near 1. Despite being a PEARL-based method, TAVT achieves significantly higher success rates, often approaching 1. Compared to on-policy algorithms like LDM, RL[2], and VariBad, TAVT also demonstrates much better performance. These results highlight the effectiveness of our method in enhancing generalization for OOD tasks. The training graphs for MetaWorld environments are provided in Appendix H.1.

To enable a more practical comparison, we report the computational cost of the proposed TAVT framework and the PEARL algorithm in Appendix H.2. While full TAVT requires approximately 18% more training time than the PEARL baseline due to the additional training of each component, other algorithms do not yield comparable performance gains even with similar computational overhead. This further demonstrates the practical advantage of TAVT.

### 5.3. Task Representation of TAVT

The comparative experiments confirm the superiority of TAVT. To explore the factors behind this improvement, Fig. 8 presents a t-SNE visualization of the task latents learned by TAVT across various environments. As shown in the cased of Ant-Goal environment in Fig. 1, TAVT ac-

curately aligns task latents for both training and OOD test tasks, reflecting task characteristics. For instance, task latents align linearly by target velocity in Cheetah-Vel-OOD and by target mass in Walker-Mass-OOD. Similarly, task latents align according to 2D target directions in Ant-Dir-4 and 3D target goals in Reach-OOD-Inter. These results demonstrate that the proposed metric-based task representation effectively distinguishes and generalizes task latents to OOD tasks. Task representations for other environments are provided in Appendix B. While t-SNE provides a reasonably meaningful visualization for multi-dimensional latents, it does not preserve exact distances. To more accurately assess whether the learned latents preserve task-wise distances under the proposed metric-based learning framework, we train TAVT with a 2D latent space and directly visualize the resulting representations. The results, presented in Appendix B.2, demonstrate that the 2D latents capture task characteristics comparably to the t-SNE projections while more faithfully preserving relative distances. This further validates the effectiveness of our metric-based representation learning approach.

### 5.4. Ablation Studies

**Component Evaluation:** To further analyze the proposed TAVT method, we perform a component evaluation on the Cheetah-Vel-OOD and Walker-Mass-OOD environments, comparing the final average return in Fig. 9. We consider the variations of TAVT including 'TAVT w/o VT', which removes virtual tasks entirely, 'TAVT w/o $\mathcal{L}_{gen}$', which excludes the $\mathcal{L}_{gen}$ component, 'TAVT w/o on-off loss', which

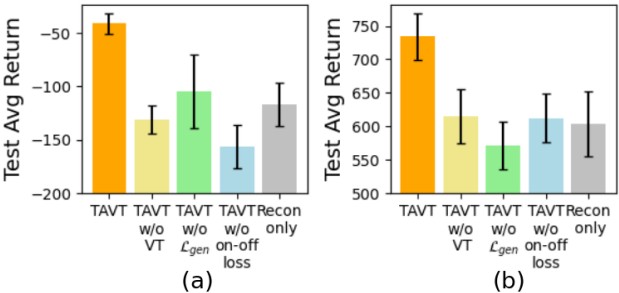

Figure 9. Component evaluation on (a) Cheetah-Vel-OOD and (b) Walker-Mass-OOD environments.

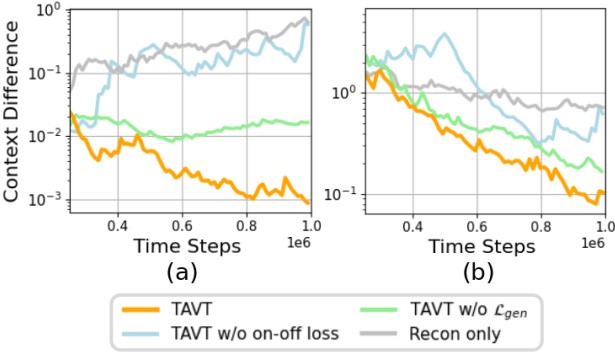

Figure 10. Context differences between real contexts and virtual contexts generated by the task decoder for OOD tasks: (a) Cheetah-Vel-OOD and (b) Walker-Mass-OOD environments.

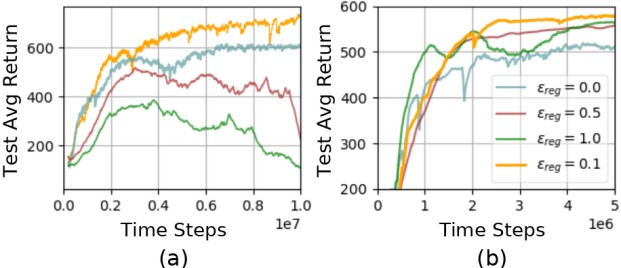

Figure 11. Performance comparison for various $\epsilon_{\text{reg}}$ on (a) Walker-Mass-OOD (b) Hopper-Mass-OOD environments.

omits the on-off loss, and 'Recon only', which uses only reconstruction loss with DropOut as in LDM. The results demonstrate that removing any component significantly degrades performance, underscoring the importance of task representation learning and sample generation for OOD task generalization. In addition, TAVT outperforms the 'Recon only' approach, showcasing its ability to construct more effective virtual tasks than existing methods.

**Context Differences:** To evaluate the impact of context differences between VT-generated and actual contexts on performance, Fig. 10 presents these differences for the TAVT variations analyzed in the component evaluation. Since 'TAVT w/o VT' does not utilize virtual contexts, it is excluded from Fig. 10. The results show that removing any component increases context error, leading to degraded per-

formance, as reflected in Fig. 9, underscoring the importance of each component in TAVT. Notably, omitting the task-preserving sample generation loss $\mathcal{L}_{\text{gen}}$ significantly increases context differences. These findings demonstrate that the proposed WGAN-based task-preserving sample generation method effectively reduces context errors and enhances generalization performance, as detailed in Section 4.2. Additional context differences for other environments are provided in Appendix C.

**State Regularization:** To demonstrate the effectiveness of the proposed state regularization method, Fig. 11 shows the performance across different $\epsilon_{\text{reg}}$ values. As illustrated, a small $\epsilon_{\text{reg}}$ effectively reduces estimation bias. Fig. 11 further evaluates its impact on performance in Walker-Mass-OOD and Hopper-Mass-OOD environments, where $\epsilon_{\text{reg}} = 0.1$ yields the best results. This setting minimizes both $Q$-function loss and overestimation bias, enabling the $Q$-function to accurately reflect expected returns and improving performance in OOD tasks. In addition, we provide a finer sweep over $\epsilon_{\text{reg}}$ on the Walker-Mass-OOD environment in Appendix C.2, where $\epsilon_{\text{reg}} = 0.1$ still yields the best performance. These findings highlight the importance of the proposed state regularization method. Additional ablation studies on the mixing coefficient $\beta$ for VT construction are provided in Appendix I.

## 6. Limitations

While TAVT shows strong performance through metric-based learning and task-aware sample generation, there are areas for improvement. First, it involves several hyperparameters that may require tuning for optimal performance, though it is not highly sensitive and our ablation studies provide practical guidance. Second, TAVT assumes a parametric task distribution (e.g., velocity or mass changes in agent dynamics) and does not cover non-parametric distributions such as those in ML10 or ML45 from MetaWorld, where task semantics differ entirely. This limitation could be addressed by combining TAVT with recent task decomposition methods (Lee et al., 2023), which we leave as a promising direction for future work.

## 7. Conclusion

Existing meta-RL methods either overlook OOD tasks or struggle with them, particularly when state transitions vary. Our proposed method, TAVT, addresses these challenges through metric-based latent learning, task-preserving sample generation, and state regularization. These components ensure precise task latents and enhanced generalization, overcoming previous limitations. Experimental results demonstrate that TAVT achieves superior alignment of OOD task latents with training task characteristics.

## Acknowledgment

This work was supported in part by the Institute of Information & Communications Technology Planning & Evaluation (IITP) grant funded by the Korea government (MSIT), under the projects: No. 2022-0-00469 (Development of Core Technologies for Task-oriented Reinforcement Learning for Commercialization of Autonomous Drones), IITP-2025-RS-2022-00156361 (Innovative Human Resource Development for Local Intellectualization), and RS-2020-II201336 (Artificial Intelligence Graduate School Support at UNIST); and in part by the 2021 Research Fund (No. 1.210149.01) of UNIST (Ulsan National Institute of Science & Technology).

## Impact Statement

This paper presents work whose goal is to advance the field of Machine Learning. There are many potential societal consequences of our work, none which we feel must be specifically highlighted here

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

## A. Proof

**Definition 4.1** (Bisimulation metric for task representation)**.** For two different tasks $\mathcal{T}_i$ and $\mathcal{T}_j$,

$$d(\mathcal{T}_i, \mathcal{T}_j) = \mathbb{E}_{(s,a)\sim D}\left[|R^{\mathcal{T}_i}(s,a) - R^{\mathcal{T}_j}(s,a)| + \eta W_2(P^{\mathcal{T}_i}(\cdot|s,a), P^{\mathcal{T}_j}(\cdot|s,a))\right], \quad (1)$$

where $D$ is the replay buffer that stores the sample contexts, $R^{\mathcal{T}}, P^{\mathcal{T}}$ are the reward function and the transition dynamics for task $\mathcal{T}$, $W_2$ is 2-Wasserstein distance between the two distributions, and $\eta \in (0, 1]$ is the distance coefficient.

**Proposition 4.2.** $d(\cdot, \cdot)$ *defined in Eq.* (2) *is a metric.*

**Proof of Proposition 4.2**

As mentioned in Section 2, each task $\mathcal{T}$ is represented by an MDP $(S, A, P^{\mathcal{T}}, R^{\mathcal{T}}, \gamma, \rho_0)$, where $S$ is the state space, $A$ is the action space, $P^{\mathcal{T}}$ represents the state transition dynamics, $R^{\mathcal{T}}$ is the reward function, $\gamma \in [0, 1)$ is the discount factor, and $\rho_0$ is the initial state distribution. Since all tasks shares the same state space and action space in this paper, so $\mathcal{T}_i = \mathcal{T}_j$ if and only if $P^{\mathcal{T}_i}(s'|s,a) = P^{\mathcal{T}_j}(s'|s,a)$ and $R^{\mathcal{T}_i}(s,a) = R^{\mathcal{T}_j}(s,a), \quad \forall s, \; s' \in S, \; a \in A$ for any tasks $\mathcal{T}_i, \; \mathcal{T}_j \in \mathcal{M}$. Thus, from the definition of $d$ given by Eq. (2), $d$ is a metric since $d$ satisfies the following axioms:

1. (Non-negativity) $d(\mathcal{T}_i, \mathcal{T}_j) \geq 0$ since $|\cdot|$ and $W_2(\cdot, \cdot)$ are non-negative,

2. $d(\mathcal{T}_i, \mathcal{T}_j) = 0 \iff R^{\mathcal{T}_i} = R^{\mathcal{T}_j}$ and $P^{\mathcal{T}_i} = P^{\mathcal{T}_j} \iff \mathcal{T}_i = \mathcal{T}_j$,

3. (Symmetry) $d(\mathcal{T}_i, \mathcal{T}_j) = d(\mathcal{T}_j, \mathcal{T}_i)$ from the definition,

4. (Triangle Inequality) $d(\mathcal{T}_i, \mathcal{T}_k) \leq d(\mathcal{T}_i, \mathcal{T}_j) + d(\mathcal{T}_j, \mathcal{T}_k)$:

$$
\begin{aligned}
d(\mathcal{T}_i, \mathcal{T}_k) &= \mathbb{E}_{(s,a)\sim D}\left[|R^{\mathcal{T}_i}(s,a) - R^{\mathcal{T}_k}(s,a)| + \eta W_2(P^{\mathcal{T}_i}(\cdot|s,a), P^{\mathcal{T}_k}(\cdot|s,a))\right], \\
&\underset{*}{\leq} \mathbb{E}_{(s,a)\sim D}\Big[|R^{\mathcal{T}_i}(s,a) - R^{\mathcal{T}_j}(s,a)| + |R^{\mathcal{T}_j}(s,a) - R^{\mathcal{T}_k}(s,a)|+ \\
&\qquad \eta W_2(P^{\mathcal{T}_i}(\cdot|s,a), P^{\mathcal{T}_j}(\cdot|s,a)) + \eta W_2(P^{\mathcal{T}_j}(\cdot|s,a), P^{\mathcal{T}_k}(\cdot|s,a))\Big], \\
&= \mathbb{E}_{(s,a)\sim D}\left[|R^{\mathcal{T}_i}(s,a) - R^{\mathcal{T}_j}(s,a)| + \eta W_2(P^{\mathcal{T}_i}(\cdot|s,a), P^{\mathcal{T}_j}(\cdot|s,a))\right]+ \\
&\qquad \mathbb{E}_{(s,a)\sim D}\left[|R^{\mathcal{T}_j}(s,a) - R^{\mathcal{T}_k}(s,a)| + \eta W_2(P^{\mathcal{T}_j}(\cdot|s,a), P^{\mathcal{T}_k}(\cdot|s,a))\right], \\
&= d(\mathcal{T}_i, \mathcal{T}_j) + d(\mathcal{T}_j, \mathcal{T}_k),
\end{aligned}
$$

where $*$ can be derived, as both the absolute value $|\cdot|$ and the Wasserstein distance $W_2$ satisfy the triangle inequality. ∎

# B. Visualization of Task Latents Representation

## B.1. Visualization of Task Latents Representation Across All Environments

To present the task latent representation results for considered OOD environments, Fig. B.1 shows the task representations for 6 MuJoCo OOD environments (Cheetah-Vel-OOD, Ant-Dir-2, Ant-Dir-4, Hopper-Mass-OOD, and Walker-Mass-OOD, and Ant-Goal-OOD), and Fig. B.2 illustrates the task representations for 4 ML1-OOD environments (Reach-OOD-Inter, Reach-OOD-Extra, Push-OOD-Inter, and Push-OOD-Extra). All task representations in Fig. B.1 and Fig. B.2 are obtained during the meta-testing phase. Additionally, to achieve stable task representations for Fig. B.2, a larger number of training tasks ($N_{\text{train}} = 500$) was sampled compared to the originally required number ($N_{\text{train}} = 50$).

From the results in Fig. B.1, the task latent representations for Cheetah-Vel-OOD, Ant-Dir-2, Ant-Dir-4, Hopper-Mass-OOD, and Walker-Mass-OOD are well-aligned according to their respective task characteristics, such as target velocity, target directions, and agent mass. For Ant-Goal-OOD, the task representation is aligned based on two characteristics: goal radius and direction. These findings demonstrate that the proposed method effectively aligns task representations with task characteristics across all MuJoCo environments, while OOD tasks maintain latents that preserve these characteristics.

Similarly, the results in Fig. B.2 show that for each ML1-OOD environment, the 3D visualization of the target goal positions is presented on the left, and the task representations aligned with the goal positions are displayed on the right. The results indicate that for all considered environments, both training and OOD tasks are well-aligned in 3D space according to their target goal positions. These findings highlight that the proposed Bisimulation metric-based task representation learning effectively creates a task latent space aligned with task characteristics, while the TAVT training method enhances the generalization of OOD task representations.

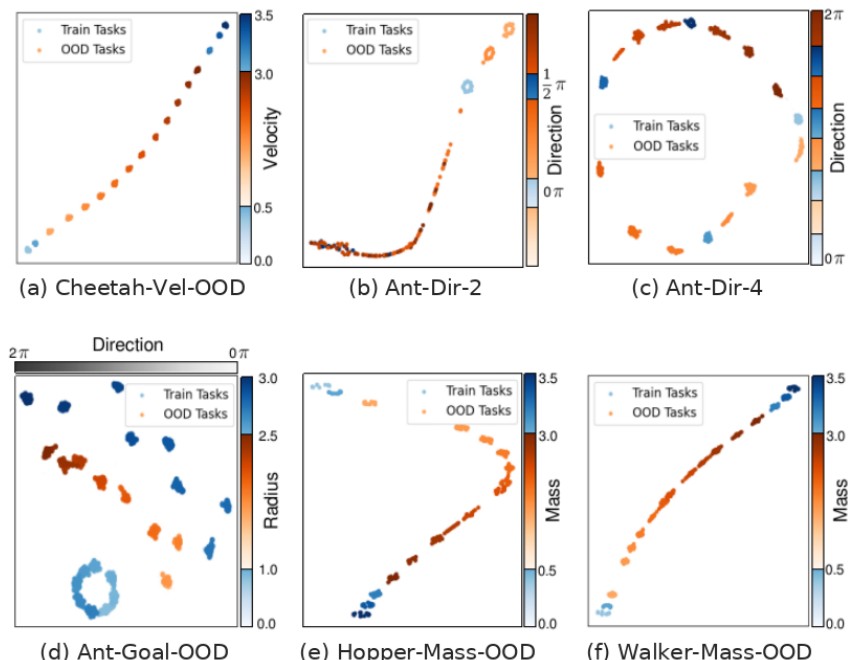

*Figure B.1.* (a-f) t-SNE visualization of the 6 MuJoCo OOD environments

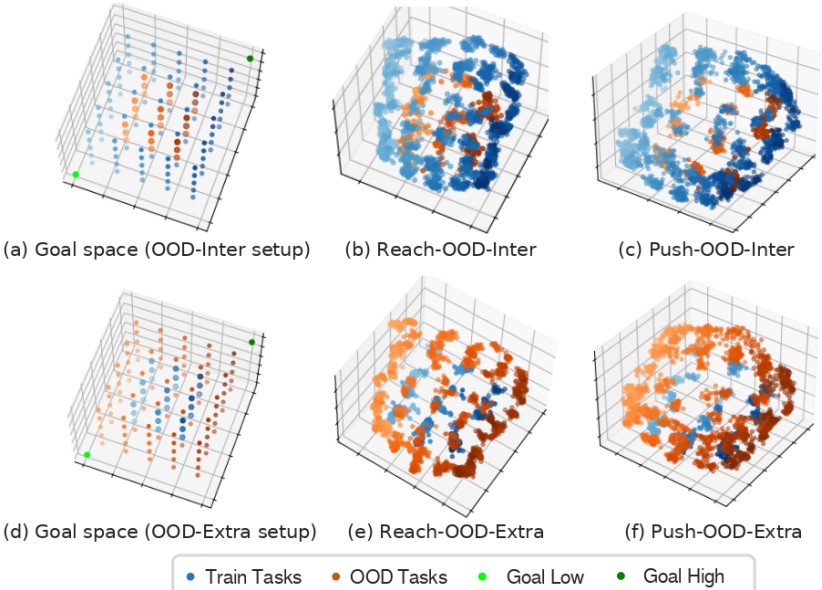

Figure B.2. (a, d) Visualization of the goal space of each ML1-OOD setup (b, c, d, f) t-SNE visualization of the 4 ML1-OOD environments

## B.2. Visualization of 2-Dimensional Task Latents Representation

While t-SNE visualization method provides a reasonably meaningful visualization for multi-dimensional latents through effective manifold learning algorithm, it does not preserve exact distances between the latent vectors. To more accurately assess whether the learned latent space actually preserves task-wise distances under the proposed metric-based learning framework, we train TAVT with a 2D latent space on Ant-Goal-OOD environment and directly visualize the resulting latent representations on 2D plane, enabling to reflect the direct distance between latent vectors. Fig. B.3(a) shows the Ant-Goal-OOD environment setup that is introduced in Fig. 1(a), and Fig. B.3(b) shows the direct visualization of 2-dimensional task latents. Each axis represents each element of 2D latent vector. Only the latent dimension changes from 10 to 2 from Fig. 1(e) setup. This direct visualization indicates the task latent space tend to generally reflect the task geometry and preserve the task-wise distance despite very low dimensional latent vector including both training tasks and OOD test tasks. This demonstrates that the 2D latents capture task characteristics comparably to the t-SNE projections while faithfully preserving relative distances and validates the effectiveness of our metric-based representation learning approach.

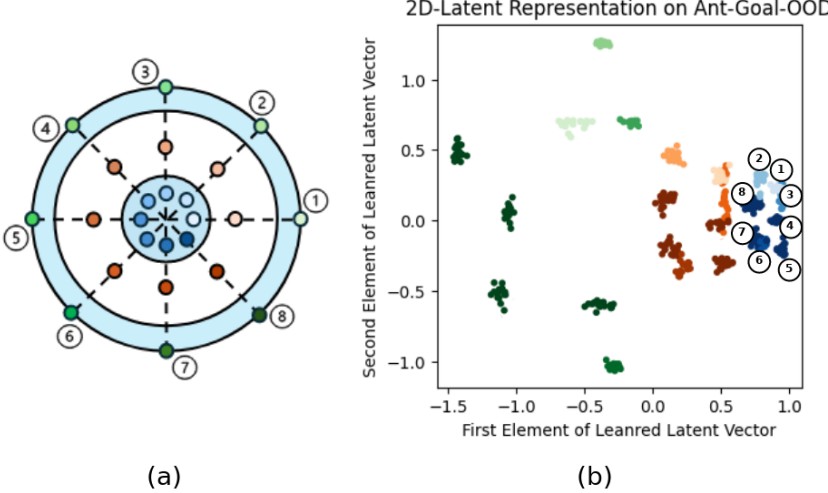

(a)                                      (b)

Figure B.3. Visualization of 2-dimensional task latent on Ant-Goal-OOD environment: (a) 2D Goal positions in the Ant-Goal-OOD environment, which is introduced in Fig. 1(a). (b) Direct visualization of 2D latent vectors on 2D plane.

# C. Ablation Studies of the Task-Preserving Sample Generation.

## C.1. Effectiveness of the Proposed Task-Preserving Sample Generation

In Section 4.2, we introduce a task-preserving sample generation technique to ensure that virtual contexts retain task latent information while closely resembling actual task samples. To evaluate the effectiveness of this approach, we analyze how virtual sample contexts generated by the task decoder differ from real task contexts for OOD tasks. Context difference graphs for all considered OOD setups are provided, showing the average reward and state differences between real and generated contexts for both MuJoCo and ML1 environments.

Fig. C.1 presents context differences across 6 MuJoCo environments under OOD setups. Our TAVT algorithm achieves the smallest context differences in all environments, with the differences being most pronounced in Cheetah-Vel-OOD, Ant-Goal-OOD, and Ant-Dir-4. Similarly, Fig. C.2 shows context differences in 4 ML1-OOD environments, where TAVT again demonstrates the smallest differences. These results highlight TAVT's effectiveness in generating accurate transition samples, even for unseen OOD tasks.

In comparison, the 'Recon only' and 'TAVT w/o on-off loss' setups exhibit the largest context differences, followed by 'TAVT w/o $\mathcal{L}_{\text{gen}}$', while the proposed TAVT method achieves the smallest context difference. The 'Recon only' method suffers due to its reliance on transition samples solely from training tasks. The absence of the proposed on-off loss or $\mathcal{L}_{\text{gen}}$ leads to increased context differences, demonstrating that these components significantly reduce errors in the sample contexts generated by the proposed VT. This reduction in context errors directly contributes to improving OOD task generalization.

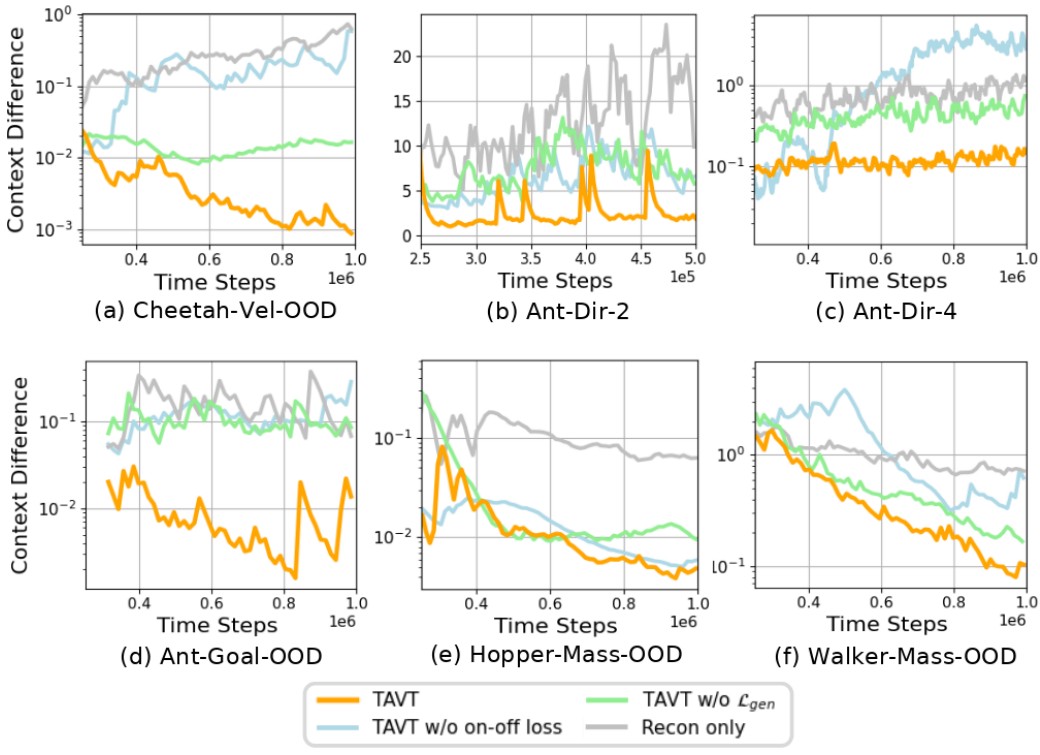

*Figure C.1.* (a-f) Context differences between real contexts and virtual contexts generated by the task decoder for the six MuJoCo OOD tasks. The $y$-axis is plotted on a log scale except Ant-Dir-2 environment to show the difference.

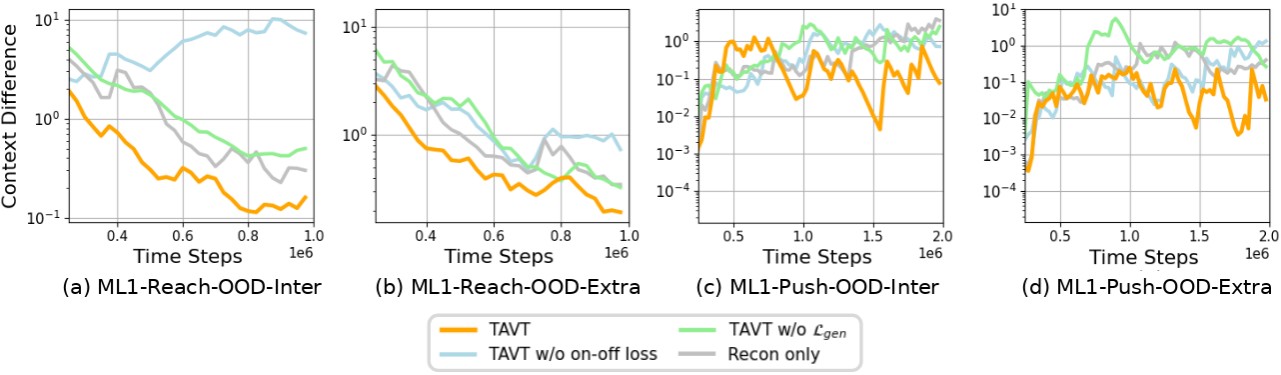

*Figure C.2.* (a-d) Context differences between real contexts and virtual contexts generated by the task decoder for the four ML1 OOD tasks. The $y$-axis is plotted on a log scale to show the difference.

### C.2. Additional Ablation study of $\epsilon_{\text{reg}}$

In Section 4.4, we propose a state regularization method to mitigate the overestimation problem that occurs when learning the $Q$-function using generated virtual states. In this section, we extend our ablation study to include a finer sweep over $\epsilon_{\text{reg}} \in [0.0, 0.05, 0.1, 0.2, 0.5, 1.0]$ on the Walker-Mass-OOD environment. Fig. C.3 shows the additional ablation experiment results for $\epsilon_{\text{reg}}$ in the Walker-Mass-OOD environment. Fig. C.3(a) presents the performance according to $\epsilon_{\text{reg}}$, exhibiting a concave trend where relatively high performance is achieved at smaller values of $\epsilon_{\text{reg}}$, with the best performance occurring at $\epsilon_{\text{reg}} = 0.1$. In addition, Fig. C.3(b) shows the estimation bias of the $Q$-function according to $\epsilon_{\text{reg}}$, defined as the difference between $Q$-function values and average actual returns. When $\epsilon_{\text{reg}} = 0.05$ or $\epsilon_{\text{reg}} = 0.1$, the estimation bias of the $Q$-function is the smallest, while the others show a greater degree of overestimation or underestimation. These results confirm that small values around $\epsilon_{\text{reg}} = 0.1$ consistently lead to lower $Q$-function estimation bias and improved return and imply that the proposed state regularization method effectively helps the $Q$-function to learn from the generated virtual states.

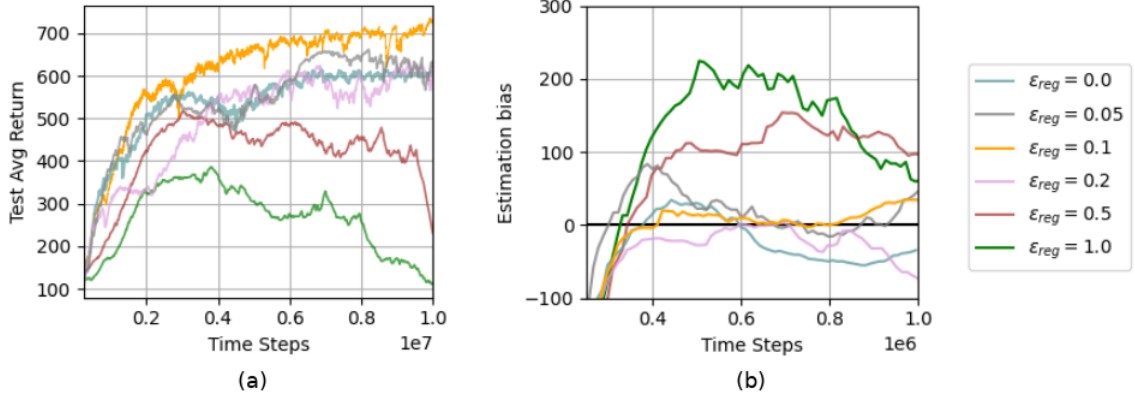

*Figure C.3.* Additional ablation study of $\epsilon_{\text{reg}}$ on Walker-Mass-OOD environment: (a) Performance comparison (b) Estimation bias for OOD tasks for various $\epsilon_{\text{reg}}$.

# D. More Detailed Implementations

In this section, we provide more detailed implementations for the proposed TAVT. Section D.1 describes the practical implementation of the loss terms in $\mathcal{L}_{\text{bisim}}$, Section D.2 explains the implementation of the gradient penalty for WGAN loss, and Section D.3 details the implementation of meta-RL with TAVT, including the meta-training and meta-testing algorithms.

## D.1. Practical Implementation of $\mathcal{L}_{\text{bisim}}$

To improve task representation and reflect task differences, we use the Bisimulation metric as described in Section 4.1. We also introduce an on-off loss to stabilize task latents. The encoder-decoder loss is given by Eq. (3), and to compute $d(\mathcal{T}_i, \mathcal{T}_j; p_{\bar{\phi}})$ in Eq. (3), we use the task decoder $p_\phi(\cdot, \bar{\mathbf{z}}_{\text{off}})$. However, since $\bar{\mathbf{z}}_{\text{off}}$ evolves during encoder training, the task decoder may become unstable, potentially affecting the metric $d$. To address this instability, we propose using an additional decoder $p_{\tilde{\phi}}(\cdot, \text{idx}_{\mathcal{T}_i})$ with parameter $\tilde{\phi}$ to compute the metric $d$, where $\text{idx}_{\mathcal{T}_i}$ is the one-hot encoded vector of the task index $i$. The decoder $p_{\tilde{\phi}}$ is also trained using reconstruction loss. Consequently, the updated encoder-decoder loss is given by

$$\mathcal{L}_{\text{bisim}}(\psi, \phi, \tilde{\phi}) = \mathbb{E}_{\mathcal{T}_i, \mathcal{T}_j \sim p(\mathcal{T}_{\text{train}})} \Big[ \underbrace{\Big( |\mathbf{z}_{\text{off}}^i - \mathbf{z}_{\text{off}}^j| - d(\mathcal{T}_i, \mathcal{T}_j; p_{\tilde{\phi}}) \Big)^2}_{\text{Bisimulation loss}} + \underbrace{\mathbb{E}_{(s,a,r,s') \sim D_{\text{off}}^{\mathcal{T}_i}, (\hat{r}, \hat{s}') \sim p_{\tilde{\phi}}(s, a, \text{idx}_{\mathcal{T}_i})} \Big[ (r - \hat{r})^2 + (s' - \hat{s}')^2 \Big]}_{\text{Reconstruction loss for } p_{\tilde{\phi}}}$$

$$+ \underbrace{\mathbb{E}_{(s,a,r,s') \sim D_{\text{off}}^{\mathcal{T}_i}, (\hat{r}, \hat{s}') \sim p_\phi(s, a, \mathbf{z}_{\text{off}}^i)} \Big[ (r - \hat{r})^2 + (s' - \hat{s}')^2 \Big]}_{\text{Reconstruction loss for } p_\phi} + \underbrace{(\mathbf{z}_{\text{on}}^i - \bar{\mathbf{z}}_{\text{off}}^i)^2}_{\text{on-off latent loss}} \Big], \quad \mathbf{z}^i \sim q_\psi(\cdot | \mathbf{c}^{\mathcal{T}_i}), \ \mathbf{c}^{\mathcal{T}_i} \sim D^{\mathcal{T}_i}, \ \forall i. \quad \text{(D.1)}$$

Moreover, to further stabilize the learning of the latent variables, instead of using a single sample of $\mathbf{z}_{\text{off}}^i$ in the on-off loss, we use the average of $\mathbf{z}_{\text{off}}^i$ from multiple contexts sampled from the buffer. This approach helps prevent fluctuations in $\mathbf{z}_{\text{off}}^i$ due to varying contexts, thereby aiding in more stable learning of the task latents.

## D.2. Implementation of Gradient Penalty in $\mathcal{L}_{\text{disc}}$

To stabilize the training of the adversarial network, the improved WGAN framework (Gulrajani et al., 2017) incorporates a gradient penalty (GP) that restricts the gradient to prevent the discriminator from learning too quickly, as described in Section 4.2. GP ensures that the discriminator satisfies the 1-Lipschitz continuity condition. In WGAN discriminator loss Eq. (4), the Gradient Penalty term is calculated by

$$\text{Gradient Penalty} = \mathbb{E}_{\mathcal{T}_i \sim p(\mathcal{T}_{\text{train}})} [(\|\nabla_{\delta_{\text{inter}}} f_\zeta(\delta_{\text{inter}})\|_2 - 1)^2],$$

where $\delta_{\text{inter}} := \delta(\mathbf{c}^{\mathcal{T}_i}, \bar{\mathbf{z}}_{\text{off}}^i) + (1 - \delta)(\hat{\mathbf{c}}^\alpha, \bar{\mathbf{z}}_{\text{off}}^\alpha)$ represents the interpolated samples between $(\mathbf{c}^{\mathcal{T}_i}, \bar{\mathbf{z}}_{\text{off}}^i)$, which are induced by training tasks $\mathcal{T}_i$, and $(\hat{\mathbf{c}}^\alpha, \bar{\mathbf{z}}_{\text{off}}^\alpha)$, which are induced by the task decoder of VT. Here, $\delta \sim \text{Unif}([0, 1])$ is the interpolation factor for the GP. In addition, the WGAN structure trains the discriminator more frequently than the generator at a 5:1 ratio, as proposed in the original WGAN paper (Gulrajani et al., 2017).

## D.3. Meta-RL with TAVT and TAVT Algorithms

As described in Section 4.3, we aim to perform meta-RL using the contexts $\mathbf{c}_{\text{off}}^i$ obtained from training task $\mathcal{T}_i$ and the virtual contexts $\hat{\mathbf{c}}^\alpha$ generated by the task decoder of the VT. For meta RL, the $Q$-function and the RL policy are defined as $Q_\theta(s, a, \mathbf{z})$ and $\pi_\theta(\cdot | s, \mathbf{z})$ with task latent $\mathbf{z}$, where $\theta$ represents the parameters of both the $Q$-function and the policy $\pi$. Based on SAC (Haarnoja et al., 2018), the RL losses for the $Q$-function and the policy are then given by:

$$\mathcal{L}_Q(\theta; \mathbf{c}, \mathbf{z}) = \mathbb{E}_{(s,a,r,s') \sim \mathbf{c}} \Big[ \Big( Q_\theta(s, a, \bar{\mathbf{z}}) - (r \cdot \lambda_{\text{rew}} + \gamma \mathbb{E}_{a' \sim \pi_\theta(\cdot | s, \bar{\mathbf{z}})} [Q_{\theta^-}(s', a', \bar{\mathbf{z}}) - \lambda_{\text{ent}} \log \pi_\theta(\cdot | s, \bar{\mathbf{z}})] \Big)^2 \Big] \quad \text{(D.2)}$$

$$\mathcal{L}_\pi(\theta; \mathbf{c}, \mathbf{z}) = \mathbb{E}_{s \sim \mathbf{c}} \Big[ D_{\text{KL}} \Big( \pi_\theta(\cdot | s, \bar{\mathbf{z}}) \, \Big\| \, \frac{\exp(Q_\theta(s, \cdot, \bar{\mathbf{z}})/\lambda_{\text{ent}})}{Z_\theta(s)} \Big) \Big], \quad \text{(D.3)}$$

where $Q_{\theta^-}$ is the target value network with parameters $\theta^-$ updated from $\theta$ using the exponential moving average (EMA), $\lambda_{\text{rew}}$ is the reward scale, $\lambda_{\text{ent}}$ is the entropy coefficient, $D_{\text{KL}}(p\|q)$ is the Kullback-Leibler divergence between two distributions $p$ and $q$, and $Z_\theta$ is the normalizing factor. Based on the proposed TAVT, we update the $Q$-function and the RL policy with training task latents $\mathbf{z}^i$ using off-policy contexts obtained from these tasks, and update the $Q$-function and the

RL policy with the latents $\mathbf{z}^\alpha$ of VT using virtual contexts generated by the task decoder, as explained in Section 4.3. The total $Q$-function loss and policy loss with TAVT are given by

$$\mathcal{L}_Q(\theta) = \mathbb{E}_{\mathcal{T}_i \sim p(\mathcal{T}_{\text{train}}), \mathbf{z}_{\text{on}}^i \sim q_\psi, \mathbf{c}_{\text{off}}^{\mathcal{T}_i} \sim D_{\text{off}}^{\mathcal{T}_i}} [\mathcal{L}_Q(\theta; \mathbf{z}_{\text{on}}^i, \mathbf{c}_{\text{off}}^{\mathcal{T}_i}) + \lambda_{\text{VT}} \mathcal{L}_Q(\theta; \mathbf{z}_{\text{on}}^\alpha, \hat{\mathbf{c}}_{\text{off}}^\alpha)] \tag{D.4}$$

$$\mathcal{L}_\pi(\theta) = \mathbb{E}_{\mathcal{T}_i \sim p(\mathcal{T}_{\text{train}}), \mathbf{z}_{\text{on}}^i \sim q_\psi, \mathbf{c}_{\text{off}}^{\mathcal{T}_i} \sim D_{\text{off}}^{\mathcal{T}_i}} [\mathcal{L}_\pi(\theta; \mathbf{z}_{\text{on}}^i, \mathbf{c}_{\text{off}}^{\mathcal{T}_i}) + \lambda_{\text{VT}} \mathcal{L}_\pi(\theta; \mathbf{z}_{\text{on}}^\alpha, \hat{\mathbf{c}}_{\text{off}}^\alpha)], \tag{D.5}$$

where $\lambda_{\text{VT}} \in [0, 1]$ is the loss coefficient for VT training. If $\lambda_{\text{VT}} = 0$, then TAVT does not utilize virtual samples at all. For generated virtual contexts $\hat{\mathbf{c}}^\alpha$, we apply the state regularization method proposed in Section 4.4 for environments with varying state dynamics such as Hopper-Mass-OOD and Walker-Mass-OOD.

Here, to obtain on-policy latents $\mathbf{z}_{\text{on}}$ for meta-RL, we sample $N_{\text{exp}}$ episodes using our exploration policy $\pi_{\text{exp}} = \pi_\theta(\cdot|s, \mathbf{z}_{\text{on}}^\alpha)$, as proposed in Section 4.3. This policy allows us to explore a broad range of tasks, as $\mathbf{z}_{\text{on}}^\alpha$ spans all interpolated areas, including both training tasks and VTs. To enhance the exploration of diverse trajectories, we regenerate $\mathbf{z}_{\text{on}}^\alpha$ every $H_{\text{freq}}$ timesteps during the exploration process. This periodic update enables the exploration policy to adapt to new tasks and explore the environment with these newly assigned tasks. An analysis of the exploration policy with respect to the sampling frequency $H_{\text{freq}}$ is provided in Appendix E.2. Finally, summarizing the contents of Section 4 and Appendix D, the proposed TAVT algorithm is outlined in Algorithm 1 (Meta-training of TAVT) and Algorithm 2 (Meta-testing of TAVT).

---

**Algorithm 1** Meta Training of TAVT

---

**Require:** The training task set $\{\mathcal{T}_j\}_{j=1,\ldots,N_{\text{train}}} \sim p(\mathcal{T}_{\text{train}})$, the task encoder $q_\psi$, the task decoders $p_\phi$ and $p_{\tilde{\phi}}$, the WGAN discriminator $f_\zeta$, the $Q$-function $Q_\theta$, and the policy $\pi_\theta$.

1: Initialize parameters $\psi, \phi, \tilde{\phi}, \zeta, \theta$ and replay buffers for all training tasks.
2: **for** epoch $k = 1, 2, \cdots$ **do**
3:     Sample $N_{\text{meta}}$ training tasks from the training task set.
4:     **for** task $\mathcal{T}_i$ in $N_{\text{meta}}$ training tasks **do**
5:         Collect contexts $\mathbf{c}_{\text{on}}^{\mathcal{T}_i}$ using the exploration policy $\pi_{\text{exp}} = \pi(\cdot|s, \mathbf{z}_{\text{on}}^\alpha)$ for $N_{\text{exp}}$ episodes.
6:         Collect contexts $\mathbf{c}_{\text{off}}^{\mathcal{T}_i}$ using the RL policy $\pi_{\text{RL}} = \pi(\cdot|s, \mathbf{z}_{\text{on}}^i)$ for $N_{\text{RL}}$ episodes, where $\mathbf{z}_{\text{on}}^i \sim q_\psi(\cdot|\mathbf{c}_{\text{on}}^{\mathcal{T}_i})$.
7:         Store contexts $\mathbf{c}_{\text{on}}^{\mathcal{T}_i}$ and $\mathbf{c}_{\text{off}}^{\mathcal{T}_i}$ in the replay buffers $D_{\text{on}}^{\mathcal{T}_i}$ and $D_{\text{off}}^{\mathcal{T}_i}$, respectively.
8:     **end for**
9:     Construct $N_{\text{VT}}$ virtual tasks using the training task latents.
10:    Generate the virtual contexts $\hat{\mathbf{c}}_{\text{off}}^\alpha$ for each VT using the task decoder $p_\phi$.
11:    **for** gradient step in $K_{\text{Model}}$ steps **do**
12:       Sample $N_{\text{meta}}$ tasks in train tasks set.
13:       Compute $\mathcal{L}_{\text{bisim}}(\psi, \phi, \tilde{\phi})$ loss by Eq. (D.1).
14:       Compute $\mathcal{L}_{\text{disc}}(\zeta)$ and $\mathcal{L}_{\text{gen}}(\psi, \phi)$ losses by Eq. (4) and Eq. (5).
15:       Update the WGAN discriminator parameter $\zeta$ by $\zeta \leftarrow \zeta - \lambda_{\text{lr}} \cdot \nabla_\zeta \mathcal{L}_{\text{disc}}(\zeta)$.
16:       Update the model parameters $\psi$ and $\phi, \tilde{\phi}$ by $(\psi, \phi, \tilde{\phi}) \leftarrow (\psi, \phi, \tilde{\phi}) - \lambda_{\text{lr,context}} \cdot \nabla_{(\psi, \phi, \tilde{\phi})} \mathcal{L}_{\text{total}}(\psi, \phi, \tilde{\phi})$.
17:    **end for**
18:    **for** gradient step in $K_{\text{RL}}$ steps **do**
19:       Compute RL losses $\mathcal{L}_Q(\theta)$ and $\mathcal{L}_\pi(\theta)$ by Eq. (D.4) and Eq. (D.5), respectively.
20:       Update the RL parameter $\theta$ by $\theta \leftarrow \theta - \lambda_{\text{lr}} \cdot \nabla_\theta(\mathcal{L}_Q(\theta) + \mathcal{L}_\pi(\theta))$.
21:    **end for**
22: **end for**

---

---

**Algorithm 2** Meta Testing of TAVT

---

**Require:** The OOD test task set $\mathcal{M}_{\text{test}}$, $\pi_\theta$, $q_\psi$.

1: **for** task $\mathcal{T}$ in $\mathcal{M}_{\text{test}}$ **do**
2:     **for** episode k=1, …, $N_{\text{exp}}$ **do**
3:         Generate $\mathbf{z}_{\text{on}}^\alpha$ every $H_{\text{freq}}$ timesteps
4:         Collect contexts $\mathbf{c}_{\text{on}}^\mathcal{T}$ using the exploration policy $\pi_{\text{exp}} = \pi(\cdot|s, \mathbf{z}_{\text{on}}^\alpha)$
5:     **end for**
6:     Task inference $\mathbf{z}_{\text{on}} \sim q_\psi(\cdot|\mathbf{c}_{\text{on}}^\mathcal{T})$.
7:     Rollout transition using $\pi_\theta(\cdot|s, \mathbf{z}_{\text{on}})$ for the last episode.
8: **end for**
9: Compute the average return of the last episodes for all test tasks.

---

## E. Exploration Trajectories of the Proposed Exploration Policy $\pi_{\text{exp}}$

In Section 4.3, we propose the exploration policy $\pi_{\text{exp}} = \pi(\cdot|s, \mathbf{z}_{\text{on}}^\alpha)$ to cover a diverse range of tasks, whereas PEARL uses $\pi(\cdot|s, \tilde{\mathbf{z}})$, $\tilde{\mathbf{z}} \sim N(\mathbf{0}, \mathbf{I})$ for exploration. We compare the exploration trajectories of PEARL and our TAVT in Section E.1. Additionally, in Section D.3, we introduce the method that update $\mathbf{z}_{\text{on}}^\alpha$ every $H_{\text{freq}}$ timesteps during the exploration phase to observe diverse trajectories, while PEARL samples $\tilde{\mathbf{z}}$ only once per episode. To demonstrate the effectiveness of $H_{\text{freq}}$, we analyze its impact on exploration diversity in Section E.2.

### E.1. Trajectory Comparison between PEARL and TAVT

The exploration process in the PEARL algorithm involves sampling $\mathbf{z}$ from a prior distribution $N(\mathbf{0}, \mathbf{I})$, as it learns to align all task latents $z$ with this prior, as discussed in Section 2. In contrast, our algorithm uses $\mathbf{z}_{\text{on}}^\alpha$ obtained from VT construction, which spans diverse regions of the latent space, including both training tasks and virtual tasks. To compare the exploration behavior of PEARL and our TAVT, Fig. E.1 and Fig. E.2 illustrate the first and second exploration trajectories during the exploration phase, as well as the final trajectory generated by $\pi_{\text{RL}}$ in the Ant-Goal-OOD environment for training tasks and OOD tasks, respectively, for (a) PEARL and (b) TAVT. From the results, it is evident that TAVT's exploration policy covers a broader range of areas in the goal space compared to PEARL's exploration policy. Since the task latent for the RL policy is selected by the task encoder based on these exploration trajectories, TAVT's ability to explore diverse goal spaces enhances the differentiation of the current task within the task latent. Consequently, TAVT successfully reaches the goal points for both training and OOD tasks due to the effectiveness of our task latent, whereas PEARL fails to achieve the goal points in both scenarios.

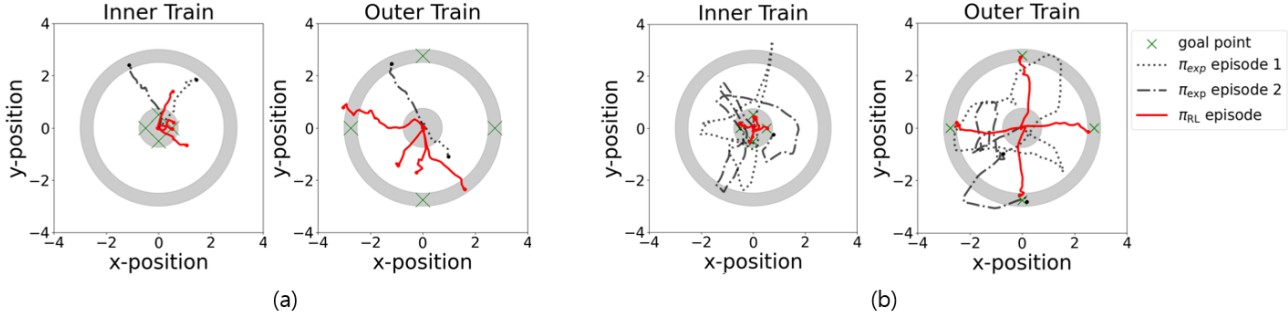

*Figure E.1.* Visualization of trajectories in the goal space during the meta-testing phase for inner and outer training tasks in the Ant-Goal-OOD environment: (a) PEARL and (b) TAVT.

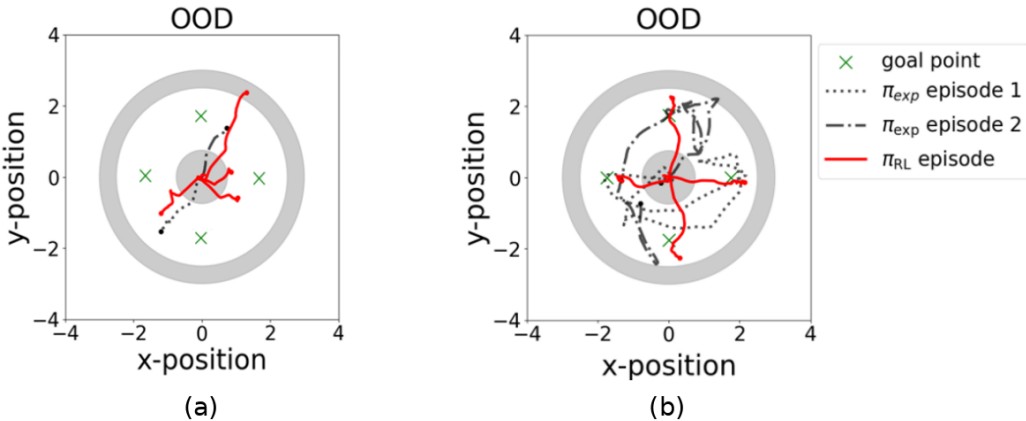

*Figure E.2.* Visualization of trajectories in the goal space during the meta-testing phase for OOD test tasks in the Ant-Goal-OOD environment: (a) PEARL and (b) TAVT.

### E.2. Exploration Trajectories with Respect to Sampling Frequency $H_{\text{freq}}$

As proposed in Section D.3, we regenerated $\mathbf{z}_{\text{on}}^{\alpha}$ every $H_{\text{freq}}$ timesteps for the proposed exploration policy $\pi_{\text{exp}}$. To demonstrate the effect of $H_{\text{freq}}$ on exploration, Fig. E.3 shows the exploration trajectories during the meta-testing phase for different values of $H_{\text{freq}}$: (a) 5, (b) 20, and (c) 200.

From Fig. E.3, it is evident that if the sampling frequency is too low, such as $H_{\text{freq}} = 5$, the exploration policy results in trajectories that are confined to the area around the starting point without covering much distance. In contrast, if the sampling frequency is too high, like $H_{\text{freq}} = 200$ (the episode horizon for Ant-Goal-OOD environment), the agent can reach locations far from the starting point but fails to explore a diverse set of directions. We found that a balanced sampling frequency, such as $H_{\text{freq}} = 20$, allows the agent to explore a wide range of directions while still covering distant areas effectively.

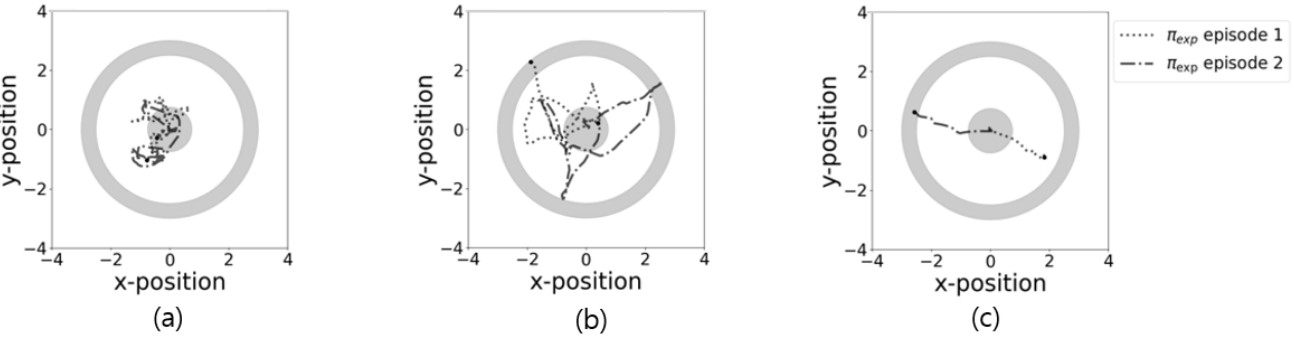

*Figure E.3.* Visualization of exploration trajectories of TAVT according to the sampling frequency $H_{\text{freq}}$: (a) 5 steps (b) 20 steps (c) 200 steps

# F. Hyperparameter Setup for TAVT

In this section, we provide a detailed hyperparameter setup for the proposed TAVT. To do this, we first define the coefficients for the loss functions related to the Bisimulation metric-based task representation and task-preserving sample generation proposed in Section 4 and Appendix D.1 as follows:

$$\mathcal{L}_{\text{bisim}}(\psi, \phi) = \mathbb{E}_{\mathcal{T}_i, \mathcal{T}_j \sim p(\mathcal{T}_{\text{train}})} \Bigg[ \lambda_{\text{bisim}} \cdot \underbrace{\left( |\mathbf{z}_{\text{off}}^i - \mathbf{z}_{\text{off}}^j| - d(\mathcal{T}_i, \mathcal{T}_j; p_{\tilde{\phi}}) \right)^2}_{\text{Bisimulation loss}}$$

$$+ \lambda_{\text{recon}} \cdot \underbrace{\mathbb{E}_{(s,a,r,s') \sim D_{\text{off}}^{\mathcal{T}_i}, (\hat{r}, \hat{s}') \sim p_{\tilde{\phi}}(s, a, \text{idx}_{\mathcal{T}_i})} \left[ (r - \hat{r})^2 + (s' - \hat{s}')^2 \right]}_{\text{Reconstruction loss for } p_{\tilde{\phi}}}$$

$$+ \lambda_{\text{recon}} \cdot \underbrace{\mathbb{E}_{(s,a,r,s') \sim D_{\text{off}}^{\mathcal{T}_i}, (\hat{r}, \hat{s}') \sim p_{\phi}(s, a, \mathbf{z}_{\text{off}}^i)} \left[ (r - \hat{r})^2 + (s' - \hat{s}')^2 \right]}_{\text{Reconstruction loss for } p_{\phi}}$$

$$+ \lambda_{\text{on-off}} \cdot \underbrace{(\mathbf{z}_{\text{on}}^i - \bar{\mathbf{z}}_{\text{off}}^i)^2}_{\text{on-off latent loss}} \Bigg], \quad \mathbf{z}^i \sim q_{\psi}(\cdot | \mathbf{c}^{\mathcal{T}_i}), \ \mathbf{c}^{\mathcal{T}_i} \sim D^{\mathcal{T}_i}, \ \forall i. \tag{F.6}$$

$$\mathcal{L}_{\text{disc}}(\zeta) = \lambda_{\text{WGAN}} \cdot \mathbb{E}_{\mathcal{T}_i \sim p(\mathcal{T}_{\text{train}}), \mathbf{c}^{\mathcal{T}_i} \sim D^{\mathcal{T}_i}} \left[ -f_{\zeta}(\mathbf{c}^{\mathcal{T}_i}, \bar{\mathbf{z}}_{\text{off}}^i) + \mathbb{E}_{\hat{\mathbf{c}}^{\alpha} \sim p_{\phi}}[f_{\zeta}(\hat{\mathbf{c}}^{\alpha}, \bar{\mathbf{z}}_{\text{off}}^{\alpha})] \right] + \lambda_{\text{GP}} \cdot \text{ Gradient Penalty}, \tag{F.7}$$

$$\mathcal{L}_{\text{gen}}(\psi, \phi) = \mathbb{E}_{\hat{\mathbf{c}}^{\alpha} \sim p_{\phi}} [ -\lambda_{\text{WGAN}} \cdot \underbrace{f_{\zeta}(\hat{\mathbf{c}}^{\alpha}, \bar{\mathbf{z}}_{\text{off}}^{\alpha})}_{\text{WGAN generator loss}} + \lambda_{\text{TP}} \cdot \underbrace{\mathbb{E}_{\hat{\mathbf{z}}^{\alpha} \sim q_{\psi}(\cdot | \hat{\mathbf{c}}^{\alpha})}[(\hat{\mathbf{z}}^{\alpha} - \bar{\mathbf{z}}_{\text{off}}^{\alpha})^2]}_{\text{task preserving loss}} ]], \tag{F.8}$$

where $\lambda_{\text{bisim}}$ is the Bisimulation loss coefficient, $\lambda_{\text{recon}}$ is the reconstruction loss coefficient, $\lambda_{\text{on-off}}$ is the on-off loss coefficient, $\lambda_{\text{WGAN}}$ is the WGAN loss coefficient, $\lambda_{\text{TP}}$ is the task-preserving loss coefficient, and $\lambda_{\text{GP}}$ is the gradient-penalty coefficient. Along with these loss coefficients, Table F.1 displays the shared hyperparameter setup across all environments, while Table F.2 and F.3 presents the hyperparameter setup specific to each environment. As shown in Tables F.1, F.2 and F.3 the proposed TAVT has more hyperparameters compared to PEARL. However, most of the loss coefficients are similar across different environments, and the environment-specific hyperparameters listed in Table F.2 and F.3 are the same as those used in PEARL. Thus, in practice, TAVT does not require a significantly larger hyperparameter search compared to PEARL.

*Table F.1.* Shared hyperparameters

| | Name | Value (for MuJoCo) | Value (for MetaWorld) |
|---|---|---|---|
| | $\lambda_{\text{bisim}}$ | 100 (50 for Cheetah-Vel-OOD) | 100 |
| | $\lambda_{\text{recon}}$ | 200 | 200 |
| | $\lambda_{\text{on-off}}$ | 100 | 100 |
| Loss coefficients | $\lambda_{\text{wgan}}$ | 1.0 | 1.0 |
| | $\lambda_{\text{preserve}}$ | 100 | 100 |
| | $\lambda_{\text{VT}}$ | 1.0 (0.1 for Hopper-Mass-OOD) | 1.0 (for Reach) / 0.1 (for Push) |
| | $\lambda_{\text{GP}}$ | 5.0 | 5.0 |
| Shared Hyper-parameters | Context learning rate $\lambda_{\text{lr,context}}$ | 0.0003 | 0.0003 |
| | Learning rate $\lambda_{\text{lr}}$ | 0.0003 | 0.0003 |
| | Optimizer | Adam | Adam |
| | Mixing coefficient $\beta$ | 2.0 | 2.0 |
| | Distance coefficient $\eta$ | 0.1 | 1 (for Reach) / 10 (for Push) |
| | Regularization coefficient $\epsilon_{\text{reg}}$ | 0.1 | - |
| | Latent dimension | 10 | 10 |
| | Batch size for RL | 256 | 512 |
| | Context batch size $N_c$ | 128 | 256 |
| | Num. of exploration trajectories $N_{\text{exp}}$ | 2 (4 for Ant-Goal-OOD) | 2 |
| | Num. of RL trajectories $N_{\text{RL}}$ | 3 (6 for Ant-Dir-4) | 3 |
| | Sampling frequency $H_{\text{freq}}$ | 20 | 50 |
| | Model gradient steps per epoch $K_{\text{model}}$ | 500 (1000 for Walker/Hopper-Mass-OOD) | 1000 |
| | RL gradient steps per epoch $K_{\text{RL}}$ | 4000 (1000 for Cheetah-Vel-OOD) | 4000 |
| | $q_\psi, Q_\theta, \pi_\theta$ | [300,300,300] | [300,300,300] |
| Network sizes | $p_\phi, p_{\tilde{\phi}}$ | [256,256,256] | [256,256,256] |
| | $f_\zeta$ | [200,200,200] | [200,200,200] |

*Table F.2.* MuJoCo environmental hyperparameters

| | Name | Environments | | | | | |
|---|---|---|---|---|---|---|---|
| | | Cheetah-Vel-OOD | Ant-Dir-2 | Ant-Dir-4 | Ant-Goal-OOD | Hopper-Mass-OOD | Walker-Mass-OOD |
| Environmental Hyperparameters | Reward scale $\lambda_{\text{rew}}$ | 5.0 | 5.0 | 5.0 | 1.0 | 5.0 | 5.0 |
| | Entropy coefficient $\lambda_{\text{ent}}$ | 1.0 | 0.5 | 0.5 | 0.5 | 0.2 | 0.2 |
| | Num. of training tasks $N_{\text{train}}$ | 100 | 2 | 4 | 150 | 100 | 100 |
| | Task batch size $N_{\text{meta}}$ | 16 | 2 | 4 | 16 | 16 | 16 |
| | Num. of VTs $N_{\text{VT}}$ | 5 | 1 | 2 | 5 | 5 | 5 |
| | Num. of mixing tasks $M$ | 3 | 2 | 2 | 3 | 3 | 3 |

*Table F.3.* MetaWorld ML1 environmental hyperparameters

| | Name | Environments | | | | | |
|---|---|---|---|---|---|---|---|
| | | Reach | Reach-OOD-Inter | Reach-OOD-Extra | Push | Push-OOD-Inter | Push-OOD-Extra |
| Environmental Hyperparameters | Reward scale $\lambda_{\text{rew}}$ | 1.0 | 1.0 | 1.0 | 5.0 | 5.0 | 5.0 |
| | Entropy coefficient $\lambda_{\text{ent}}$ | 0.2 | 0.2 | 0.2 | 1 | 1 | 1 |
| | Num. of training tasks $N_{\text{train}}$ | 50 | 50 | 50 | 50 | 50 | 50 |
| | Task batch size $N_{\text{meta}}$ | 16 | 16 | 16 | 16 | 16 | 16 |
| | Num. of VTs $N_{\text{VT}}$ | 5 | 5 | 5 | 5 | 5 | 5 |
| | Num. of mixing tasks $M$ | 3 | 3 | 3 | 3 | 3 | 3 |

# G. More Detailed Experimental Setups

In this section, we provide details about the experiments and experimental setup. In G.1, we describe the OOD setup of the MuJoCo environments; in G.2, we explain the OOD setup of the ML1 environments; in G.3, we outline the baseline setup.

## G.1. Mujoco Environments

In this section, we provide a detailed description of the computational setup and the MuJoCo environments considered in Section 5. We utilized Mujoco environments from the OpenAI Gym library (Brockman et al., 2016) and employed MuJoCo 200 libraries for environments with varying reward functions (Cheetah-Vel-OOD, Ant-Dir-2, Ant-Dir-4, Ant-Goal-OOD). For environments with different state transition dynamics (Hopper-Mass-OOD, Walker-Mass-OOD), we used MuJoCo 131 libraries, as suggested in our baseline implementation algorithm, PEARL (Rakelly et al., 2019). In all of Mujoco environments, we used a dense reward setup, which remains the same as the setup used in previous studies (Finn et al., 2017; Rakelly et al., 2019; Zintgraf et al., 2019).

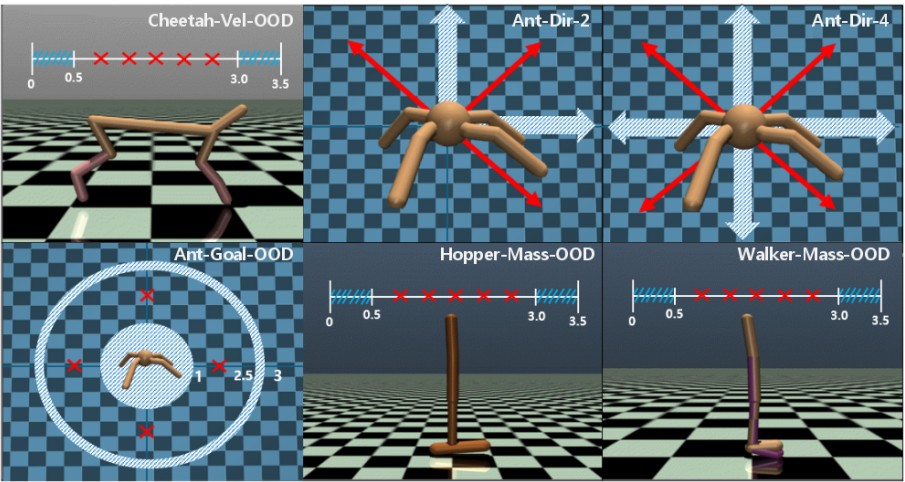

*Figure G.1.* Task configurations for the MuJoCo environments under consideration are presented. Training tasks are indicated in blue, while OOD test tasks are marked in red.

To provide a more detailed description of the Mujoco environments, Fig. G.1 illustrates the task configurations for the environments considered. In addition, for each Mujoco environment, we explain the design of the reward function, how components of the environment change with different tasks, and the configuration of out-of-distribution OOD tasks as follows:

- Cheetah-Vel-OOD: The Cheetah agent is tasked with moving at target velocities $v_{\text{tar}}$ where the reward increases as the agent's velocity more closely matches the target. The reward function is designed to be the negative value of the difference between the current velocity and the target velocity. For training tasks, target velocities $v_{\text{tar}}$ are sampled from the training task space $\mathcal{M}_{\text{train}} = [0.0, 0.5) \cup [3.0, 3.5)$. For OOD tasks, target velocities are sampled from the test task space $\mathcal{M}_{\text{test}} = \{0.75, 1.25, 1.75, 2.25, 2.75\}$.

- Ant-Dir-2: The Ant agent is required to move in target directions $\theta_{\text{dir}}$, with higher rewards given when the agent's movement aligns more accurately with the target direction. The reward function is the dot product of the agent's velocity and the target direction. During training, target directions $\theta_{\text{dir}}$ are sampled from the training task space $\mathcal{M}_{\text{train}} = \{0, \frac{\pi}{2}\}$. For OOD tasks, target directions are sampled from the test task space $\mathcal{M}_{\text{test}} = \{\frac{\pi}{4}, \frac{3\pi}{4}, \frac{7\pi}{4}\}$, including extrapolated tasks ($\frac{3\pi}{4}$ and $\frac{7\pi}{4}$ directions) from the training task space.

- Ant-Dir-4: The Ant agent is required to move in target directions $\theta_{\text{dir}}$, similar to the Ant-Dir-2 environment. The key difference lies in the task set. In this environment, the target directions $\theta_{\text{dir}}$ for training tasks are sampled from the training task space $\mathcal{M}_{\text{train}} = \{0, \frac{\pi}{2}, \pi, \frac{3\pi}{2}\}$. For OOD tasks, the target directions are sampled from the test task space $\mathcal{M}_{\text{test}} = \{\frac{\pi}{4}, \frac{3\pi}{4}, \frac{5\pi}{4}, \frac{7\pi}{4}\}$.

- Ant-Goal-OOD: The Ant agent is tasked with reaching specific 2D goal positions given by $(r_{\text{goal}} \cos \theta_{\text{goal}}, r_{\text{goal}} \sin \theta_{\text{goal}})$. The agent receives a higher reward for reaching the goal position more accurately. The goal reward function is designed as the negative $L_1$ distance from the current position to the target goal position. For training, the goal positions $r_{\text{goal}}$ and $\theta_{\text{goal}}$ are sampled from the training task space $\mathcal{M}_{\text{train}}$ with $r_{\text{goal}} \in [0.0, 1.0) \cup [2.5, 3.0)$ and $\theta_{\text{goal}} \in [0, 2\pi]$. For OOD test tasks, the goal positions are sampled from the test task space $\mathcal{M}_{\text{test}}$ with $r_{\text{goal}} = 1.75$ and $\theta_{\text{goal}} \in \{0, \frac{\pi}{2}, \pi, \frac{3\pi}{2}\}$.

- Hopper-Mass-OOD: The Hopper agent is required to move forward while dealing with varying body mass across different tasks, which alters the state transition dynamics. The reward function used is the same as that in the original MuJoCo Hopper environment. The task set is constructed by adjusting the initial mass of all joints of the Hopper using a multiplier $m_{\text{scale}}$, which is an internal parameter of the environment that scales the mass up or down. For training, the body mass multiplier $m_{\text{scale}}$ is sampled from the training task space $\mathcal{M}_{\text{train}} = [0.0, 0.5) \cup [3.0, 3.5)$. For OOD test tasks, the body mass multiplier is sampled from the test task space $\mathcal{M}_{\text{test}} = \{0.75, 1.25, 1.75, 2.25, 2.75\}$.

- Walker-Mass-OOD: The Walker2D agent is tasked with moving forward while managing varying body mass across different tasks, which affects the state transition dynamics. The reward function remains consistent with that used in the original MuJoCo Walker2D environment. The task set is created by adjusting the initial mass of all joints of the Walker using a multiplier $m_{\text{scale}}$, an internal parameter of the environment that scales the mass either up or down, as similar to Hopper-Mass-OOD. For training, the body mass multiplier $m_{\text{scale}}$ is sampled from the training task space $\mathcal{M}_{\text{train}} = [0.0, 0.5) \cup [3.0, 3.5)$. For OOD test tasks, the body mass multiplier is sampled from the test task space $\mathcal{M}_{\text{test}} = \{0.75, 1.25, 1.75, 2.25, 2.75\}$.

### G.2. MetaWorld ML1 Environments

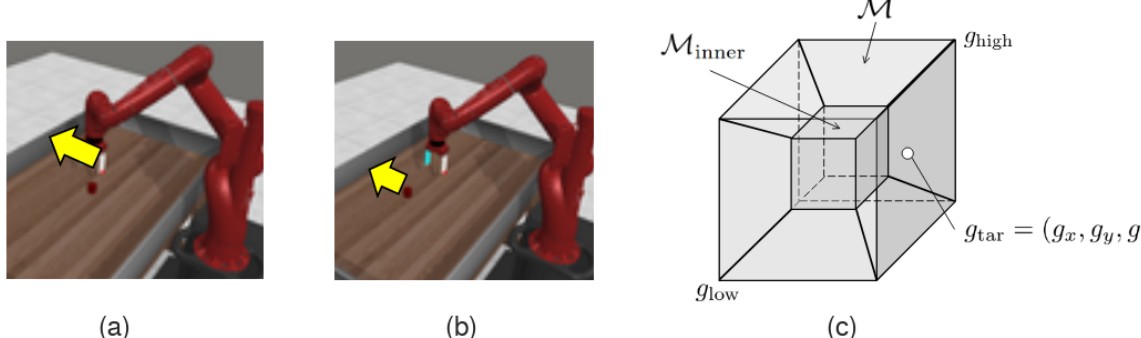

*Figure G.2.* (a) Reach environment that the end effector should reach to target position. (b) Push environment that the robot arm should push the object to the target position. (c) Visualization of ML1-OOD setup that separate 3D goal space $\mathcal{M}$ into $\mathcal{M}_{\text{inner}}$ and $\mathcal{M} \backslash \mathcal{M}_{\text{inner}}$.

ML1-Reach and ML1-Push are environments in the MetaWorld benchmark (Yu et al., 2020), where a robotic arm either reaches a target position in 3D space (ML1-Reach) or pushes an object to a target position (ML1-Push). Tasks are defined by the goal target position, $g_{\text{tar}}$, and their reward functions are determined accordingly. The target position is sampled from a 3D goal space $\mathcal{M}$, a rectangular cuboid defined by two vertices, $g_{\text{low}}$ and $g_{\text{high}}$, which are given in the original environment code and differ between ML1-Reach and ML1-Push. Training and test tasks are sampled from $\mathcal{M}_{\text{train}}$ and $\mathcal{M}_{\text{test}}$ with $N_{\text{train}}$ and $N_{\text{test}}$ tasks, respectively. In the basic ML1 setup, $\mathcal{M}_{\text{train}}$ and $\mathcal{M}_{\text{test}}$ are identical to $\mathcal{M}$.

For OOD setups, we introduce a smaller central region, $\mathcal{M}_{\text{inner}}$ that is defined within $\mathcal{M}$. The goal space $\mathcal{M}$ is divided into 125 smaller cuboids ($5 \times 5 \times 5$), with $\mathcal{M}_{\text{inner}}$ consisting of 27 central cuboids ($3 \times 3 \times 3$). Fig. G.2(a) represents the Reach environment, Fig. G.2(b) shows the Push environment, and Fig. G.2(c) illustrates the goal configuration in the MetaWorld environments. Table G.1 summarizes the OOD setups for ML1 environments.

- Basic ML1 setups: The training goal target positions (train tasks) are sampled with 50 positions from the regions of $\mathcal{M}$, while the testing goal target positions (test tasks) are sampled with 50 positions from the same regions of $\mathcal{M}$.

- OOD-Inter setup: The training goal target positions (train tasks) are sampled with 50 positions from the regions of

$\mathcal{M}$ excluding $\mathcal{M}_{\mathrm{inner}}$, while the testing goal target positions (test tasks) are composed of the center points of the 27 cuboids within $\mathcal{M}_{\mathrm{inner}}$.

- OOD-Extra setup: The training goal target positions (train tasks) are sampled with 50 positions from within $\mathcal{M}_{\mathrm{inner}}$, while the testing goal target positions (test tasks) are composed of the center points of the remaining 98 $(125 - 27)$ cuboids outside $\mathcal{M}_{\mathrm{inner}}$.

*Table G.1.* Configuration of basic ML1 and ML1 OOD environments setup.

| | $g_{\mathrm{low}}$ | $g_{\mathrm{high}}$ | $\mathcal{M}_{\mathrm{train}}$ | $\mathcal{M}_{\mathrm{test}}$ | $N_{\mathrm{train}}$ | $N_{\mathrm{test}}$ |
|---|---|---|---|---|---|---|
| Reach | | | $\mathcal{M}$ | $\mathcal{M}$ | 50 | 50 |
| Reach-OOD-Inter | $(-0.1, 0.8, 0.05)$ | $(0.1, 0.9, 0.3)$ | $\mathcal{M}\backslash\mathcal{M}_{\mathrm{inner}}$ | $\{27 \text{ center points of } \mathcal{M}_{\mathrm{inner}}\}$ | 50 | 27 |
| Reach-OOD-Extra | | | $\mathcal{M}_{\mathrm{inner}}$ | $\{98 \text{ center points of } \mathcal{M}\backslash\mathcal{M}_{\mathrm{inner}}\}$ | 50 | 98 |
| Push | | | $\mathcal{M}$ | $\mathcal{M}$ | 50 | 50 |
| Push-OOD-Inter | $(-0.1, 0.8, 0.01)$ | $(0.1, 0.9, 0.02)$ | $\mathcal{M}\backslash\mathcal{M}_{\mathrm{inner}}$ | $27$ center points of $\mathcal{M}_{\mathrm{inner}}$ | 50 | 27 |
| Push-OOD-Extra | | | $\mathcal{M}_{\mathrm{inner}}$ | $\{98 \text{ center points of } \mathcal{M}\backslash\mathcal{M}_{\mathrm{inner}}\}$ | 50 | 98 |

### G.3. Baselines Implementation

**RL[2]**  We utilize the open source codebase of LDM at https://github.com/suyoung-lee/LDM for MuJoCo environments and open source codebase of garage https://github.com/rlworkgroup/garage for MetaWorld ML1 environments to report the results of RL[2]. We modify the task space of MuJoCo and ML1 environments to be divided into $\mathcal{M}_{\mathrm{train}}$ and $\mathcal{M}_{\mathrm{test}}$, in other words, OOD setup.

**VariBAD and LDM**  We utilize the open source reference of LDM at https://github.com/suyoung-lee/LDM for MuJoCo environments and open source reference of SDVT https://github.com/suyoung-lee/SDVT for MetaWorld ML1 environments to acquire the results of VariBAD and LDM. We modify the task space of MuJoCo and ML1 environments to be divided into $\mathcal{M}_{\mathrm{train}}$ and $\mathcal{M}_{\mathrm{test}}$.

**MAML**  We utilize the open source code of garage at https://github.com/rlworkgroup/garage for both MuJoCo and ML1 environments to get the results of MAML. We modify the task space of MuJoCo and ML1 environments to be divided into $\mathcal{M}_{\mathrm{train}}$ and $\mathcal{M}_{\mathrm{test}}$.

**PEARL**  We utilize the open source repository of PEARL at https://github.com/katerakelly/oyster for MuJoCo environments and the open source repository of garage https://github.com/rlworkgroup/garage for ML1 environments to get the results of PEARL. We modify the task space of MuJoCo and ML1 environments to be divided into $\mathcal{M}_{\mathrm{train}}$ and $\mathcal{M}_{\mathrm{test}}$.

**CCM, MIER, Amago**  We utilize the open source code of CCM, MIER and Amago at https://github.com/TJU-DRL-LAB/self-supervised-rl/tree/ece95621b8c49f154f96cf7d395b95362a3b3d4e/RL_with_Environment_Representation/ccm, https://github.com/russellmendonca/mier_public and https://github.com/UT-Austin-RPL/amago, respectively for both MuJoCo and ML1 environments to measure the experimental results of the baselines. We modify the task space of MuJoCo and ML1 environments to be divided into $\mathcal{M}_{\mathrm{train}}$ and $\mathcal{M}_{\mathrm{test}}$.

**TAVT**  We research and develop TAVT algorithm on top of the PEARL official open-source algorithm at https://github.com/katerakelly/oyster for both MuJoCo environments and ML1 environments. Our implementation code is available at https://github.com/JM-Kim-94/tavt.git

We use the MetaWorld benchmark open source at https://github.com/Farama-Foundation/Metaworld for all experiments, but install the "paper version" of it. All experiments are conducted on a GPU server with an NVIDIA GeForce RTX 3090 GPU and AMD EPYC 7513 32-Core processors running Ubuntu 20.04, using PyTorch.

# H. Additional Results for TAVT

## H.1. Learning Curves for MetaWorld ML1 Environments

Fig. H.1 presents the learning curves of the 6 ML1 environments from the comparison experiments in Section 5. Similar to the results in Table 3, the learning curves show that the proposed TAVT consistently outperforms other on-policy and off-policy meta-RL methods. Notably, compared to off-policy algorithms, TAVT demonstrates both superior convergence performance and faster convergence speed. These results further highlight the effectiveness and superiority of the proposed TAVT algorithm.

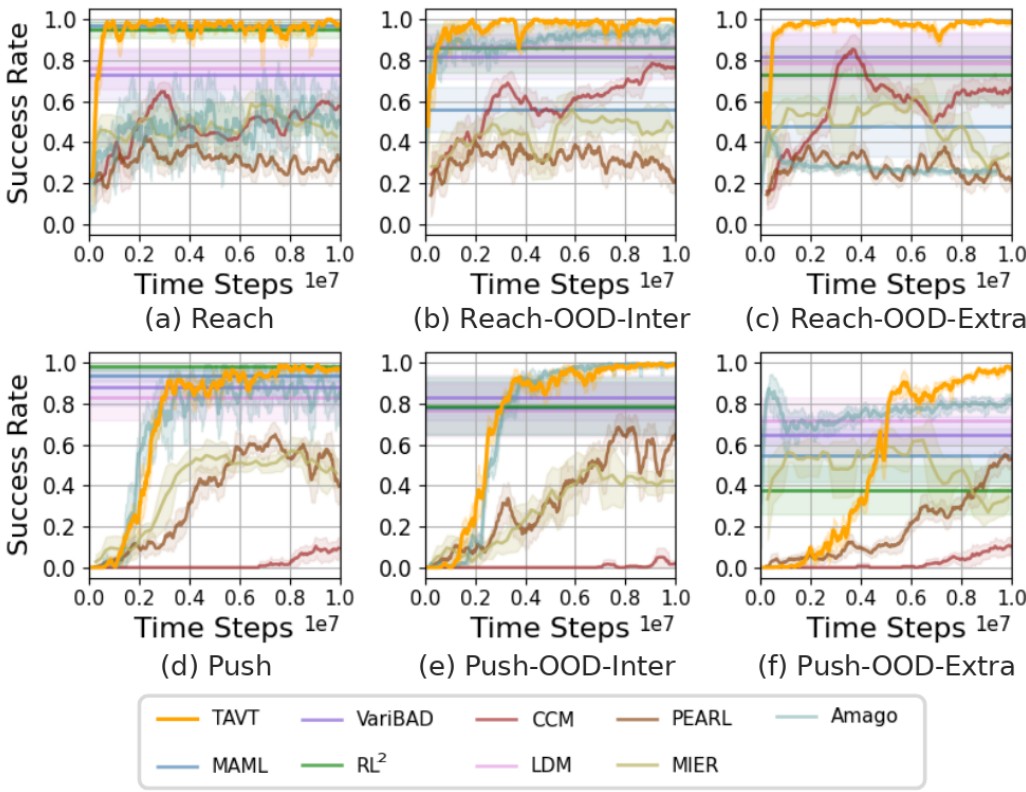

*Figure H.1.* (a-f) Performance comparison of the 6 MetaWorld ML1 environments

## H.2. Computational Cost

We provide a breakdown of the relative computational cost introduced by each component in TAVT in the same setup. We report in Table H.1 the relative training time per epoch that occurs when each component of TAVT is added compared to the PEARL algorithm we base our implementation on. As shown in Table H.1, adding the reconstruction loss and decoder contributes approximately 3% overhead compared to PEARL, while the metric-based representation (w/o $\mathcal{L}_{gen}$) adds about 11%. While full TAVT requires approximately 18% more training time than the PEARL baseline due to the additional training of each component, other algorithms do not yield comparable performance gains even with similar computational overhead, demonstrating the practical advantage of TAVT. TAVT significantly outperforms all baselines across diverse OOD environments in MuJoCo and MetaWorld, as shown in Fig. 7. Even with extended training, other methods do not reach TAVT's level of generalization. We therefore believe the added cost is reasonable and practical.

*Table H.1.* Relative per-epoch training time comparison over component variants

| Method | PEARL | Recon only | TAVT w/o $\mathcal{L}_{gen}$ | TAVT w/o on-off loss | TAVT(full) |
|---|---|---|---|---|---|
| **Relative Training Time per Epoch** | 100% | 103% | 111% | 116% | 118% |

## I. Ablation Study on the Mixing Coefficient $\beta$

In this section, we discuss the impact of the mixing coefficient $\beta$ for VT construction introduced in Section 2. Recall that the task latent for VT, denoted as $\mathbf{z}^\alpha$, is obtained by interpolating among $M$ randomly sampled training task latents, as follows:

$$\mathbf{z}^\alpha = \sum_{i=1}^{M} \alpha^i \mathbf{z}^i,$$

where $\alpha = (\alpha^1, \cdots, \alpha^M) \sim \beta\text{Dirichlet}(1, 1, \ldots, 1) - \frac{\beta-1}{M}$ is the interpolation coefficient. Here, Dirichlet$(\cdot)$ represents the Dirichlet distribution, and $\beta \geq 1$ is the mixing coefficient. When $\beta = 1$, the interpolation is limited to within the space of the training task latents. In contrast, $\beta > 1$ enables extrapolation beyond the original task latents, allowing for a broader range of virtual tasks to be generated. To analyze the impact of the mixing coefficient $\beta$ on VT construction and learning performance, we examine the coverage range of $\mathbf{z}^\alpha$ with varying $\beta$ in Section I.1, demonstrate how exploration behavior changes with different $\beta$ values in Section I.2, and evaluate the performance based on different $\beta$ values in Section I.3.

### I.1. Coverage of $\mathbf{z}^\alpha$ with Various $\beta$

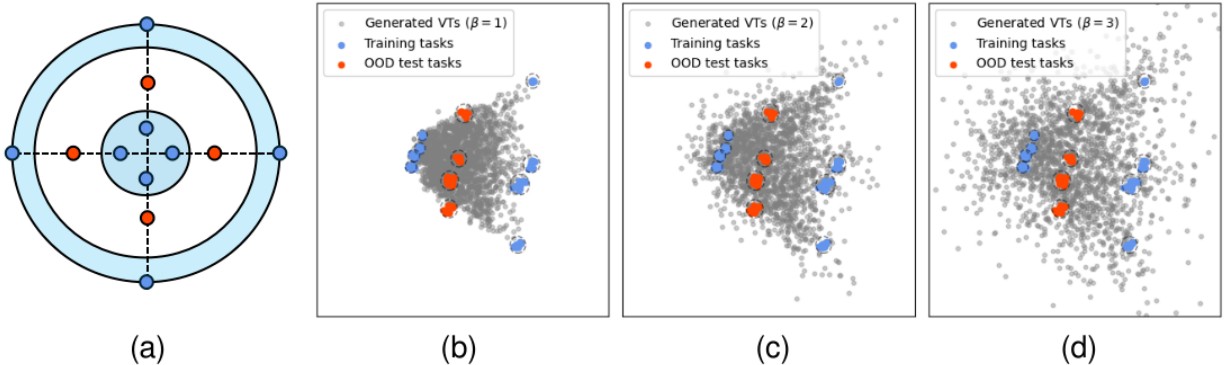

Figure I.1. (a) 2D goal positions of the Ant-Goal-OOD environment, and PCA visualizations of $\mathbf{z}^\alpha$ with different $\beta$ values: (b) TAVT with $\beta = 1$, (c) TAVT with $\beta = 2$, and (d) TAVT with $\beta = 3$.

Fig. I.1 illustrates how the coverage of task latents $\mathbf{z}^\alpha$ for VTs generated from the Dirichlet distribution varies with the mixing coefficient $\beta$ in the Ant-Goal-OOD environment. Specifically, Fig. I.1(a) displays the 2D goal positions for training and OOD tasks, while Figs. I.1(b), (c), and (d) show the principal component analysis (PCA) visualizations of $\mathbf{z}^\alpha$ generated with $\beta = 1, 2$, and 3, respectively. Here, we use PCA instead of t-SNE for visualization, as PCA more accurately reflects the data structure in the projection, whereas t-SNE often clusters data by performing manifold learning. The results in Fig. I.1 show that as $\beta$ increases, the range of task latents $\mathbf{z}^\alpha$ generated by VT construction becomes broader, effectively covering a wider area, including extrapolated task latents as intended in Section 2.

### I.2. Exploration Trajectories of $\pi_{\exp}$ with Various $\beta$

Fig. I.2 illustrates how exploration trajectories change with different $\beta$ values during the meta-testing phase. For $\beta = 1$ in Fig. I.2(a), $\mathbf{z}^\alpha$ considers only a narrow range of the task latent space, causing the agent to explore only the areas close to the inner region. For $\beta = 2$ in Fig. I.2(b), $\mathbf{z}^\alpha$ covers a broader range, which is sufficient to distinguish the tasks from the contexts. For $\beta = 3$ in Fig. I.2(c), $\mathbf{z}^\alpha$ covers the largest area but includes excessive extrapolated regions, causing the agent to explore beyond the goal space and making task distinction more challenging. Thus, we use $\beta = 2$ as the default mixing coefficient for TAVT.

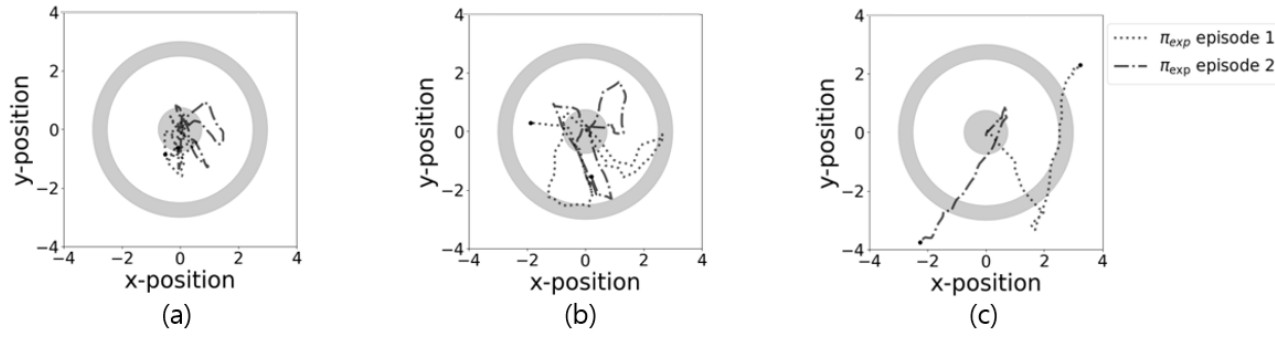

*Figure I.2.* Exploration trajectories of $\pi_{\exp}(\cdot|s, \mathbf{z}^{\alpha})$ with different $\beta$: (a) $\beta = 1$, (b) $\beta = 2$, and (c) $\beta = 3$.

### I.3. Performance Comparison with Various $\beta$

From the results in Sections I.1 and I.2, we observed that the choice of $\beta$ significantly impacts TAVT learning. Finally, to evaluate the effect of $\beta$ on performance, Fig. I.3 illustrates the performance changes across 3 environments, (a) Cheetah-Vel-OOD, (b) Ant-Goal-OOD, and (c) Walker-Mass-OOD, with varying $\beta$ values. From the results in Fig. I.3, we observe that $\beta = 2$ consistently performs the best across the considered environments. Therefore, we set $\beta = 2$ as the default mixing coefficient for TAVT.

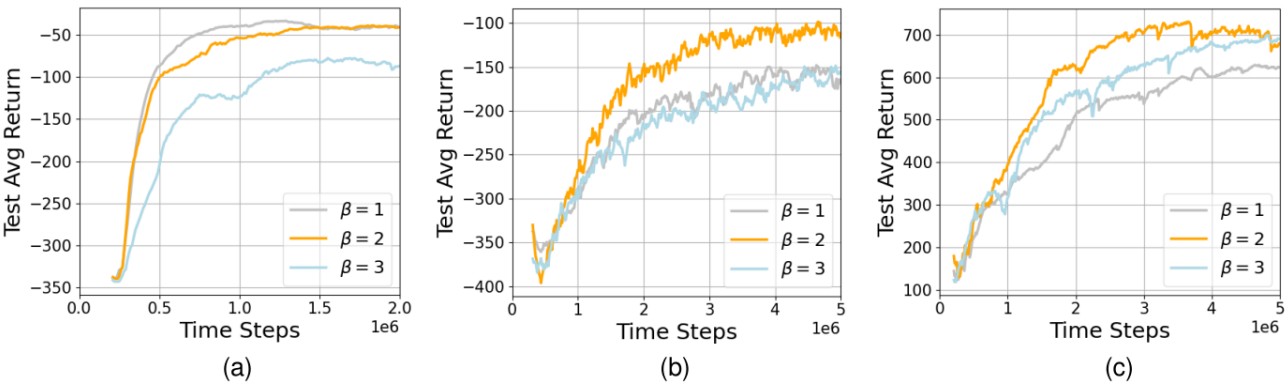

*Figure I.3.* Performance comparison with various $\beta$ values on: (a) Cheetah-Vel-OOD environment, (b) Ant-Goal-OOD environment, and (c) Walker-Mass-OOD environment.

