# OpenReview forum: "Task-Aware Virtual Training: Enhancing Generalization in Meta-Reinforcement Learning for Out-of-Distribution Tasks"
_ICML.cc/2025/Conference — ICML 2025 poster_

### Official Review · Reviewer_t99j · 2025-02-18

**Overall Recommendation:** 4

**Summary:**

This paper tackles meta RL, where the goal is find optimal policies for MDPs that have the same state-action space but different transition and reward functions. This paper is similar to prior works like PEARL but additionally uses extra loss terms/objectives to improve performance. Their main ideas are:
- Virtual tasks generation as linear combinations of observed training tasks
- They propose a distance metric for tasks and train their latent representations of tasks to reflect this distance (bisimulation loss)
- They train latents for the same tasks to be consistent whether the policy is exploratory on optimizing for that task (on-off)
- Reconstruction loss for generated samples
- Task Preserving sample generation: They train the encoder-decoder to be consistent in the sense that the generated samples from some latent z^alpha are encoded to some value hat(z)^alpha that is close to z^alpha.
- They use a GAN based approach to ensure the generated samples are indistinguishable from real samples
- They regularize state generation with some real next states
- They train a policy on both real and virtual tasks.

**Claims And Evidence:**

The claims are effectively that all of the above techniques are necessary and performant. Their claims are empirically validated, where ablations show that performance significantly degrades if any technique is removed.

**Essential References Not Discussed:**

None

**Experimental Designs Or Analyses:**

The experiments are standard for Meta RL settings. Using prior implementations for baselines is good practice.

**Methods And Evaluation Criteria:**

Yes, the proposed approach and techniques are well-motivated given the problem.

**Other Comments Or Suggestions:**

Minor:
- Extra comma on line 84, right side.
- Typo line 104
- Comma line 99 right side
- Comma 246
- There may be some typos in appendix d1 relating to phi - and phi ~. Perhaps use a different symbol then phi ~ since phi - can be interpreted as phi with no gradients blocked.

**Other Strengths And Weaknesses:**

Strengths:
- The method is thoroughly ablated
- The empirical results are on reasonably challenging benchmarks
- Each individual contribution is well-justified.

Weaknesses:
- This paper does not sufficiently address its limitations. Primarily, I am concerned about compute time. This paper introduces many new networks and loss terms, which greatly increases resource usage. In the appendix, the authors claim that their algorithms has better convergent performance. However, it would be more convincing if ablations with compute time on the x axis were provided to compare performance between approaches.
- The state regularization parameter epsilon is concerning to me for two reasons. 1) In some MDPs, taking a linear combination of states may not be sensible. For example, in a maze, two states may be feasible but their linear combination may be inside a wall. Alternatively, if the state is an image, linear combinations do not make sense. 2) The ablation of epsilon raises many questions. The ablation makes it seem like a small epsilon around 0.1 is best, with degrading performance is epsilon = 0 or 1. Furthermore, the algorithm is very sensitive to this choice. I would like to see a more thorough ablation on epsilon at a higher granularity to see what exactly is going on here. Perhaps there is some quadratic relationship, but there is not enough information at the moment to conclude this in my opinion.
- T-sne is notoriously fickle and occasionally misleading. I would be curious to see other visualization techniques to see if the expected relationships in the latent space are still preserved. For example, you could set the latent dimension to 2 so that it can be directly plotted, and see if anything interesting happens. This may degrade performance but is usually a good smell test. This is not a major issue, but is worth considering.

**Questions For Authors:**

1) In figure 6, it seems like many of the baselines have fixed performance, such as LDM, Varibad, RL^2. Why is this?
2) Does the extra decoder in section D.1, which has a one hot encoding as input, cause some scaling issues? Please discuss how well this scales since one hot encodings are not generalizable, and whether you need a one hot for only training tasks or also test tasks.

Congrats on a good paper. I am willing to increase my score if questions and weaknesses are sufficiently addressed.

**Relation To Broader Scientific Literature:**

This paper introduces novel contributions to prior meta RL algorithms, and I consider it an improved version of PEARL.  It also leverages ideas from GANS, VAEs, bisimulation, etc.

**Theoretical Claims:**

I've checked the theoretical claims/proof in appendix A. It appears correct, but I believe there are some almost everywhere arguments that need to be addressed for completeness. The proposed metric may be a pseudo metric, though this has little practical implication.

---

> ### Author Rebuttal · Authors · 2025-03-31
>
> We sincerely thank the reviewer for their thoughtful and constructive comments. We would like to address the main concerns raised, particularly regarding (1) computational cost, (2) the validity of state regularizaion method, and (3) t-SNE -based visualization.
>
> **1. Computational Cost:** (common response with Reviewer q6Z7)
>
> We appreciate the reviewer’s concern regarding the potential overhead of TAVT. While direct comparison of training time across all baselines is difficult  (e.g., training time on the x-axis), we analyze the relative overhead of each TAVT component in a consistent setup. As this is a shared concern and in line with ICML’s policy encouraging shared responses, we kindly refer to **our detailed answer to Reviewer q6Z7**, which includes a full analysis. We thank for your understanding and believe it adequately addresses the concern.
>
> **2. Validity and Sensitivity of the State Regularization Method:**
>
> We agree that linear state blending may be problematic in discrete or non-Euclidean environments (e.g., image-based or maze tasks). In such cases, an alternative is to use a discriminator-guided filtering strategy, as in recent data augmentation works [1], to retain only realistic generated states. This would improve applicability to more complex domains and is a promising direction for future work. In our study, however, we focused on continuous settings, where such blending is a reasonable approximation due to smooth and Euclidean state dynamics. We will also make sure to clarify this assumption in the revised version.
>
> In addition, to address the reviewer’s insightful comment on $\epsilon_{\mathrm{reg}}$ sensitivity, we extended our ablation study to include a finer sweep over $\epsilon_{\mathrm{reg}} \in [0.0, 0.05, 0.1, 0.2, 0.5, 1.0]$ on the Walker-Mass-OOD environment. The results, shown in Table R.2, confirm that small values around $0.1$ consistently lead to lower $Q$-function estimation bias and improved return, while both extremes ($0.0$ or $1.0$) result in performance degradation. This suggests a convex relationship and supports the reviewer’s observation that a regularization sweet spot exists.
>
> **Table R.2:** Ablation on $\epsilon_{\mathrm{reg}}$ in Walker-Mass-OOD
> |$\epsilon_{\mathrm{reg}}$|0.0|0.05|0.1|0.2|0.5|1.0|
> |-|-|-|-|-|-|-|
> |Avg. Q Bias|-24.1±25.6|91.4±21.2|**36.4**±23.5|65.1±26.8|98.4±36.4|73.1±32.6|
> |Avg. Return|612.6±27.2|660.8±23.4|**745.6**±42.3|632.3±91.7|238.2±98.3|163.8±96.4|
>
> [1] Zhang et al., "Discriminator-guided model-based offline imitation learning," CoRL 2023.
>
> **3. Use of t-SNE and Latent Space Visualization:**
>
> We thank the reviewer for the valuable suggestion regarding the limitations of t-SNE and the use of alternative visualization techniques. While t-SNE is a common tool for qualitatively assessing representation structure, we agree that it can be sensitive to hyperparameters and may not always preserve the true geometry of the latent space.
>
> To address this, we additionally visualized task latents by directly setting the latent dimension to 2 in the Ant-Goal-OOD environment, thereby avoiding t-SNE entirely. The resulting plots (linked below) show raw 2D latent representations without projection, and the alignment with task goal positions confirms that our method preserves semantic structure even without dimensionality reduction artifacts.
>
> Link: https://anonymous.4open.science/r/024/ant-goal-ood-2D.PNG
>
> We will include these results in the revised version and appreciate the reviewer’s suggestion, which strengthened the empirical support for our metric-based task representation method.
>
> **4. Minor Points and Additional Questions:**
>
> We sincerely thank the reviewer for their detailed reading and thoughtful suggestions. We address the raised points as follows and will incorporate the necessary revisions in the updated version:
>
> - Proposition 3.1: While the theoretical metric is defined via true expectations, our practical implementation relies on samples from learned transition models. As such, minor approximation errors may occur, particularly in unvisited regions of the state space across tasks. We will clarify this limitation.
>
> - Typos and notation issues: We appreciate the reviewer’s careful attention and will correct all identified typos and symbols in the revision.
>
> - Figure 6: On-policy methods generally require significantly more samples for convergence. Following common practice in meta-RL, we report their final performance when comparing with off-policy methods.
>
> - One-hot encoding: The one-hot decoder is used only for training tasks. While scalability could be a concern with an extremely large number of training tasks (usually not large due to memory constraints), this has no impact on test-time generalization.
>
> Once again, we sincerely thank the reviewer for their constructive feedback and positive recommendation. We believe these clarifications will further improve the clarity and completeness of our work.

---

> > ### Comment · Reviewer_t99j · 2025-04-01
> >
> > 1. Awesome
> > 2. Interesting. Glad to see its relatively smooth.
> > 3. Thanks for running this ablation. Its interesting that the learned representation is split left to right by distance, rather than in rings spreading out from the origin or something similar.
> >
> > Increasing my score to accept.

---

> > > ### Author Response · Authors · 2025-04-07
> > >
> > > Dear Reviewer,
> > > Thank you very much for your kind and encouraging evaluation. We truly appreciate your thoughtful feedback and are grateful for your generous assessment of our work. Thank you again for your time and support.

---

### Official Review · Reviewer_q6Z7 · 2025-02-27

**Overall Recommendation:** 3

**Summary:**

In context-based meta RL, an agent is trained to assign a context vector to an MDP. It learns to identify the context, and to take actions (policy) based on it. This paper investigates the setting where an agent is tested in an MDP not seen during training.
Several ideas are introduced to improve generalization to unseen MDPs:
1. Using a bisimulation metric idea, contexts of bi-similar MDPs are trained to be similar.
2. The agent is trained on imagined data from contexts that are "mixes" of seen contexts.
3. A GAN is used to learn better reward+next state predictions for the imagined contexts.
In experiments, agents are evaluated on MDPs that were not trained on. The variations in the MDPs are all  low-dimensional - a single velocity direction, a 2/3 dimensional goal, a single mass coefficient, etc. The proposed method is shown to outperform previous methods.

## Post rebuttal:
The authors' rebuttal clarified the comaprison with Mendonca et al., 2020 (MIER) and mostly addressed my main concern of comparison with baselines tailored for OOD meta RL. From the list of references the authors mention, Ajay et al., 2022 also seems like a relevant (and more recent) comparison, but the paper already includes a rather extensive experimental evaluation.

Overall, and after reading the other reviews, I'm more convinced that this is an incremental but solid contribution to OOD tasks in Meta RL, and is a good fit for ICML.

I have changed my score to weak accept.

**Claims And Evidence:**

The experiments are thorough, and the paper is well written. The main idea makes sense, and extensive experiments support it (but see my question about missing baselines later).

One issue that should be emphasized: the task distributions here are all low-dimensional, and I suspect that this is a limitation of the method (like it is for other context-based meta RL).

**Essential References Not Discussed:**

A relevant theoretical investigation of generalization in meta RL, and how virtual tasks can help generalization appears in [1] and should be mentioned. That paper also explicitly discussed the dimension of the task space, and why such methods are limited to low dimensional tasks.

[1] Rimon, Z., Tamar, A., & Adler, G. (2022). Meta reinforcement learning with finite training tasks-a density estimation approach. Advances in Neural Information Processing Systems, 35, 13640-13653.

**Experimental Designs Or Analyses:**

In high level the experiments follow standard meta RL evaluations and look OK.

**Methods And Evaluation Criteria:**

The authors experiment with several benchmark tasks that are standard in meta RL, and the comaprison makes sense.

It would be good if the authors explicitly discuss the reward functions used. I had to look in the code to verify that dense rewards were used. Dense rewards make some domain (like ant goal) much easier, but this is a choice that has been popularized in several prior works.

I was missing baselines that explicitly account for distribution shift, except for LDM. Citing from the related work section: "some studies tackle distributional shift challenges through robust learning (Mendonca et al., 2020; Mehta et al., 2020; Ajay et al., 2022; Greenberg et al., 2023)". Why are these methods not compared with?

**Other Comments Or Suggestions:**

It may be worthwhile to discuss some future work. How can some of the ideas here be used beyond this specific application, or what problems are still missing in the current solution.

**Other Strengths And Weaknesses:**

The paper is very well written, easy to follow, and the appendix provides extensive details. Also, code is provided (although I did not run it).

This paper has a rather narrow scope: how to technically improve meta RL by improving the virtual tasks the agent is trained on. Within this scope, the authors have done a good job at coming up with several good ideas, and extensively evaluated their approach. The overall method is relatively complex (includes training a GAN, several loss terms, etc), but since the authors provide code, I guess that only time will tell if it gets picked up and used in more practical applications (beyond the toy benchmarks).

Overall, if it wasn't for my reservations about the missing baselines, I would have proposed to accept this paper, as it makes an incremental but clear contribution to meta RL.

**Questions For Authors:**

Why is there no empirical comparison with methods that explicitly account for distribution shift?

**Relation To Broader Scientific Literature:**

The paper contribution is relatively narrow: within the scope of meta-RL with low-dimensional task distribution, it proposes several technical advances for improving generalization. The idea of training on virtual tasks has been proposed before, but the technical improvements proposed here make sense and are reported to work well.

The authors nicely discuss other meta RL studies on OOD generalization in the related owrk section, but these are mostly missing from the empirical evaluation.

**Theoretical Claims:**

The (fairly simple) claim on the bi-simulation metric being a metric looks correct.

---

> ### Author Rebuttal · Authors · 2025-03-31
>
> We sincerely thank the reviewer for their thoughtful and constructive comments. We would like to address the main concerns raised, particularly regarding (1) comparison with baselines handling distributional shift, (2) the scope and practicality of our method, (3) clarification on reward functions, (4) the connection to theoretical work on generalization, and (5) additional future work.
>
> **1. Comparison with Baselines Addressing Distributional Shift:**
>
> We appreciate the reviewer’s insightful suggestion to compare with prior work that explicitly targets distributional shift. We would like to clarify that among the baselines the reviewer mentioned, MIER (Mendonca et al., 2020) is indeed already included in our empirical evaluation. We selected MIER as it is a PEARL-based method specifically designed to handle distributional shift, closely aligned with our setting, and it provides open-source code for reproducibility. Additionally, as the reviewer noted, LDM is also an algorithm that addresses OOD test tasks through virtual task generation, and is therefore included in our comparative evaluation.  We regret that the citation for Mendonca et al. (2020) was inadvertently omitted, and we will correct this in the revised version to avoid confusion.
>
> Moreover, we designed our comparison to cover a diverse range of meta-RL algorithms across categories, including both on-policy and off-policy methods (e.g., VariBad, RL$^2$, PEARL), methods with advanced task representations (e.g., CCM using contrastive learning, Amago using transformer-based in-context learning), as well as those employing virtual tasks (e.g., LDM). As a result, we focused on representative or most relevant algorithms per category rather than exhaustively including all prior works, due to experimental feasibility.
>
> **2. Scope and Practicality of the Method:**
>
> We acknowledge and appreciate the reviewer’s comment on the scope of our work. While our approach lies within the context of meta-RL with virtual task training, we argue that it has broader practical implications. One of the core contributions of TAVT is its ability to generalize across a wide spectrum of task shifts, including:
> State transition variations (e.g., in Walker-Mass/Hopper-Mass), reward variations (e.g., Ant-Dir/Cheetah-Vel), and 3D task variation (ML1-Push/Reach).
>
> By addressing these diverse sources of task variability, TAVT takes a step closer toward real-world applicability, where test tasks often differ substantially from those seen during training. We agree that practical validation in real robotic systems or high-dimensional continuous control domains remains a promising future direction, and we will add a discussion of this in the revised version.
>
> **3. Clarification on Reward Functions:**
>
> Thank you for highlighting the need to clarify the reward functions used in our experiments. We confirm that dense rewards were used in all environments, following standard practice in recent meta-RL literature. For example, Ant-Goal-OOD uses dense distance-based rewards as in PEARL and LDM. We will explicitly clarify these reward formulations in the main text and provide detailed specifications in Appendix G to ensure transparency.
>
> **4. Connection to Theoretical Work on Generalization:**
>
> We thank the reviewer for pointing out the connection to Rimon et al. (2022), which presents a theoretical framework for understanding generalization in meta-RL via density estimation. Their work highlights the challenge of generalizing to low-density regions in the task distribution. In contrast, our method complements this by densifying the latent task space through virtual task generation and bisimulation-aligned interpolation. This allows the agent to train on transitions that approximate those in unseen test tasks.
>
> We see this connection as a valuable opportunity for synergy: combining our sample generation strategy with density-aware task weighting could lead to more principled and efficient task coverage, especially under limited task availability. We will cite and discuss this work in the revised paper and consider integrating its ideas in future research.
>
> **5. Additional Future Work:**
>
> We appreciate the reviewer’s encouragement to explore broader extensions. Beyond the theoretical connection mentioned above, one promising direction is the application of TAVT to offline meta-RL. In offline settings, OOD task generalization is especially challenging due to limited and biased data. We believe that our VT-based sample generation can augment offline datasets with behaviorally consistent transitions, helping mitigate the distributional gap between training and test tasks. We plan to explore this extension in future work.
>
> Once again, we thank the reviewer for their detailed feedback and thoughtful suggestions. We hope our clarifications address the raised concerns and help situate our contribution within the broader meta-RL literature.

---

### Official Review · Reviewer_4UJT · 2025-03-13

**Overall Recommendation:** 2

**Summary:**

This paper introduces Task-Aware Virtual Training (TAVT), a novel meta-reinforcement learning algorithm designed to enhance generalization to out-of-distribution (OOD) tasks.  The numerical results demonstrate that TAVT significantly enhances generalization to OOD tasks across various MuJoCo and Meta-World environments.

**Claims And Evidence:**

The paper claims that TAVT enhances generalization to OOD tasks.  This claim is supported by numerical results in MuJoCo and Meta-World environments.  The results consistently show TAVT outperforming other meta-RL methods in OOD settings.

**Essential References Not Discussed:**

No

**Experimental Designs Or Analyses:**

The experimental design involves comparing TAVT with other meta-RL methods in MuJoCo and MetaWorld environments.  The authors analyze the task representation of TAVT and conduct ablation studies to evaluate the contribution of different components of TAVT.

**Methods And Evaluation Criteria:**

The method applies several mechanisms, including metric-based task representation, data generation, and state regularization to improve the OOD generalization ability of context-based RL. The proposed methods somehow make sense.

The evaluation criteria include comparing TAVT with various on-policy and off-policy meta-RL methods across MuJoCo and MetaWorld environments.  The performance is measured by the average return on OOD test tasks.

**Other Comments Or Suggestions:**

MetaWorld is a good benchmark with OOD test tasks. The most important feature is the heterogeneous types of tasks in ML10 and ML45. However, these scenarios are not tested by the paper.

**Other Strengths And Weaknesses:**

Strengths

The paper is well-structured and clearly written.

The paper addresses an important problem in meta-RL: generalization to OOD tasks.

The paper proposes a novel and effective algorithm, TAVT, that outperforms existing methods.

The paper provides comprehensive experimental results and analysis to support its claims.

Weaknesses

The method applies several mechanisms, including metric-based task representation, data generation, and state regularization to improve the OOD generalization ability of context-based RL. The proposed methods somehow make sense. However, the mechanisms are not novel.  For example, the data generation and data mix-up are widely applied to improve the generalization. The bisim loss in metric-based task representation is similar to the combination of the contrastive loss and reconstruction loss, which has been widely discussed in context-based RL papers.

The proposed method generates virtual task samples using GAN, which may increase the training cost of context-based RL.

**Questions For Authors:**

In the bisim loss (3), is the computation cost of the task distance metric high?

**Relation To Broader Scientific Literature:**

The method applies several mechanisms, including metric-based task representation, data generation, and state regularization to improve the OOD generalization ability of context-based RL.

**Theoretical Claims:**

Yes

---

> ### Author Rebuttal · Authors · 2025-03-31
>
> We sincerely thank the reviewer for their thoughtful and constructive comments. We would like to address the main concerns raised, particularly regarding (1) the novelty of our method, (2) benchmark coverage, and (3) computational cost.
>
> **1. Novelty of Our Approach:**
>
> We appreciate the reviewer’s comment on novelty. To address the concern, we clarify that our method is not a generic combination of task representation and VT, but a principled framework designed to answer a central question: **how can we generate virtual tasks that are truly task-aware**, that is, faithful to underlying task dynamics and semantics?
>
> To this end, we propose a metric-based task representation that aligns task latents using the Bisimulation metric. Unlike contrastive methods that separate positive and negative pairs without modeling precise distances, the Bisimulation metric captures behavioral similarity through reward and transition dynamics. This enables the latent space to reflect task geometry, supporting meaningful interpolation and extrapolation. We also introduce an on-off alignment loss to enforce consistency between off-policy latents (used in training) and on-policy latents (used in meta-testing), which is essential for stable generalization. As shown in Fig. 1, our method yields more structured and consistent task representations compared to contrastive-learning-based methods like CCM.
>
> Building on this aligned space, we present a task-preserving sample generation strategy using WGAN, coupled with a latent consistency constraint: generated virtual contexts should reconstruct their source latent when re-encoded. This ensures that generated data remains both realistic and task-consistent. As demonstrated in Fig. 3, our method effectively captures OOD task semantics, in contrast to prior VT-based methods that suffer from latent mismatch.
>
> Together, these two components form the core of TAVT. We believe this task-aware design is the key to its strong OOD generalization, supported by our visualizations and ablations. While representation learning is a widely explored area in meta-RL, it remains a crucial pillar for generalization. Our work extends this line by explicitly aligning representations with task semantics during both training and deployment.
>
> **2. Benchmark Scope:**
> We appreciate the reviewer’s thoughtful suggestion regarding ML10 and ML45. These benchmarks feature nonparametric task variation (e.g., changes in object types or task semantics), which differ fundamentally from the parametric setups we target, such as variations in velocity, mass, or 3D goal positions of tasks. In parametric cases, virtual tasks can be generated meaningfully through latent interpolation, making VT-based augmentation both tractable and effective. In contrast, ML10/45 often require task decomposition or modular policies due to their discrete and heterogeneous structure.
>
> For example, SDVT [1] tackles this by clustering tasks through task decomposition and applying per-cluster VT modules using LDM, rather than building a unified latent space. Accordingly, our method can be seen as complementary: integrating our task-aware VT generation within each sub-task cluster in nonparametric settings could enhance generalization. We will include this as a promising future direction in the revised version.
>
> [1] Lee et al., "Parameterizing non-parametric meta-reinforcement learning tasks via subtask decomposition," NeurIPS 2023.
>
> **3. Computational Cost:** (common response with Reviewer t99j)
>
> We appreciate the reviewers’ concern regarding the potential computational overhead of TAVT. While it is difficult to directly compare training time across all on-policy and off-policy baselines, we provide a breakdown of the relative cost introduced by each component in TAVT in the same setup. As shown in Table R.1, adding the reconstruction loss and decoder contributes approximately 3% overhead compared to PEARL, while the metric-based representation (w/o $\mathcal{L}_{\text{gen}}$) adds about 11%. When the full TAVT is considered (using WGAN), the total overhead reaches approximately 18%.
>
> Despite this increase, TAVT significantly outperforms all baselines across diverse OOD environments in MuJoCo and MetaWorld, as shown in Fig. 6. Even with extended training, other methods do not reach TAVT’s level of generalization. We therefore believe the added cost is reasonable and practical. This analysis will be added to the revised version for improved clarity.
>
> **Table R.1:** Relative Per-Epoch Training Time Across Component Variants
> |Method|Relative Training Time per Epoch|
> |-|-|
> |PEARL|100%|
> |Recon only|103%|
> |TAVT w/o $\mathcal{L}_{\text{gen}}$|111%|
> |TAVT w/o on-off loss|116%|
> |TAVT (full)|118%|
>
> Once again, we thank the reviewer for their detailed feedback and insightful suggestions. We hope our clarifications help address the raised concerns and further highlight the contribution of our work within the broader meta-RL literature.

---

### Official Review · Reviewer_vcob · 2025-03-14

**Overall Recommendation:** 4

**Summary:**

This paper provides a new method for meta Reinforcement Learning that focuses on generalizing to novel tasks. The paper provides a method for generating virtual tasks that takes in to account task characteristics. A metric-based task representation is learned using the Bisimulation metric. Furthermore task preserving sample generation is used to generate additional samples generalize to unseen tasks. With a combination of these, Task-Aware Virtual Training, trains a task latent dependent policy using Soft-Actor Critic. The paper compares the new method against several baselines across many different environments.

**Claims And Evidence:**

- A claim is made that the task their task-preserving loss ensures that the virtual contexts better retain task latent information. If the meaning of this is that there is parity between the virtual latent and the output of the encoder to the virtual context, then I believe that Figure 3 makes a reasonable case for this claim.

- A claim is made that their exploration policy covers a _wider_ range of trajectories. This was compared with PEARL. The evidence in Appendix E shows for the Ant-Goal problem what trajectories from exploration policies trajectories look like after the first and second episodes looked like. I think that this claim is visually shown in the Figures.

- A claim is made that the proposed method out-performs the compared baselines, as well as that generalization is good. I believe that the results from Figure 6 and Table 3 show case this.

**Essential References Not Discussed:**

None to my knowledge.

**Experimental Designs Or Analyses:**

- It is unclear to me how many repeats of each experiment were carried out to obtain the results in Figure 6 and Table 3.

- Am I right in understanding that the averages in Figure 6 for example are also across the different test tasks (for each environment)?

**Methods And Evaluation Criteria:**

Yes.

**Other Comments Or Suggestions:**

### Introduction
1. LDM is used before writing its expansion
2. Figure 1: Can a description of what the annotation (1) -> (8) represents in Figure 1(e) be written in the caption.
3. 'Despite its benefits, we identify _several_ key issues with the existing methods for VT construction' -> Since you identify two issues, wouldn't it be more prudent to say '[...] identify _two_ key issues [...].'?

### Section 2
4. After the definition of the task encoder loss, there is an extra comma.
5. End of Page 2: typo 'aallows'
6. End of Page 2: You mention a 'task decoder' for the first time as if it was mentioned previously in the section. Could you describe the 'task decoder' in the first paragraph of Section 2.2 please?

**Other Strengths And Weaknesses:**

### Strengths
- Good literature review
- Generally written well
- Good experimental comparisons across many different method from the literature.
- Appendix F and G details a lot of the experimental setup. I believe this is sufficient for reproducibility.
- Good ablation studies are carried out.
### Weaknesses
- I had trouble understanding the algorithm described in words in Section 4.3 until I looked at the Algorithm 1 in the Appendix. Perhaps the steps of the algorithm can be explained step-by-step (by breaking into a numbered list for example).
- Limitations are discussed in the Appendix. Perhaps this should be done in the main text.

**Questions For Authors:**

### Introduction
1. 'First, generated VTs often fail to capture task characteristics accurately.' -> What is the justification for this? Is this from your own analysis or from the literature. If the latter, please give a reference.
2. Figure 1:  Is there a notion in which there would be equivalent points (1) -> (8) in (b) - (d)?

### Section 2
3. What do you mean by 'allows extrapolation beyond the original latents'?

#### Section 4
4. line 193: What do you mean by '[...] virtual contexts generalize to unseen tasks.'. Do you mean that the generated contexts would correspond to real 'unseen' tasks? Or that the task latent that would correspond to the generated contexts would be close to virtual context?

**Relation To Broader Scientific Literature:**

This paper does a good literature review, and also carries out comparisons to other meta learning algorithms. The paper builds on top of methods from Latent Dynamic Mixtures [1], and also uses the Bisimulation metric [2]

[1] Lee et al, Improving generalization in meta-rl with imaginary tasks from latent dynamics mixture. 2021
[2] Zhang et al, Learning invariant representations for reinforcement learning without reconstruction. 2021

**Theoretical Claims:**

I checked the Proof of Proposition 3.1. I believe it is correct.

---

> ### Author Rebuttal · Authors · 2025-03-31
>
> We sincerely thank the reviewer for their thoughtful and constructive comments. We would like to address the main concerns raised, particularly regarding (1) issues of clarity, and (2) technical questions.
>
> **1. Clarity Improvements in Weaknesses Part:**
>
> We sincerely thank the reviewer for pointing out aspects of the paper that could benefit from improved clarity.
>
> - Section 4.3 Algorithm Description: We appreciate the feedback that the algorithmic workflow in Section 4.3 may be difficult to follow in its current narrative form. To improve readability, we will revise this section to include a clearer step-by-step breakdown, either by adding a numbered list or directly referencing steps from Algorithm 1 in the appendix.
>
> - Limitations in Appendix: We agree that the discussion of limitations is important and should be accessible to all readers. We will relocate the key points from the appendix into the main body of the paper, allowing them to be more visible and impactful.
>
> **2. Technical Clarifications for Questions:**
>
> Thank you for the thoughtful questions. Below are our clarifications, which will also be reflected in the revised manuscript:
>
> - Averaging in Figure 6 and Table 3: All performance metrics in these figures are averaged over 5 random seeds. For each seed, the reported value reflects the average performance across all considered OOD test tasks in that environment. The shaded areas and error bars represent the standard deviation across these seeds. We will explicitly clarify this in both the main text and figure captions to avoid any ambiguity.
>
> - Equations D.4 and D.5 (Source of $\mathbf{z}^\alpha$ and $\hat{\mathbf{c}}^\alpha$): The virtual latent $\mathbf{z}^\alpha$ is computed through linear interpolation between training task latents as described in Section 2.3. The virtual context $\hat{\mathbf{c}}^\alpha$ is then generated using the task decoder conditioned on $\mathbf{z}^\alpha$ as described in Section 4.2. We will revise the appendix to clearly connect these definitions across sections.
>
> - "First, generated VTs often fail to capture task characteristics accurately": We would like to clarify that the statement was not referencing a claim made in prior literature, but rather reflects our own empirical observation based on the results shown representation of LDM in Figure 1. Specifically, although LDM also utilizes VT generation, we observe that its task representations do not align well with task semantics such as goal direction in the Ant-Goal environment. We will revise the manuscript to clearly indicate this.
>
> - Figure 1 (1)→(8) Annotation: The indices (1) through (8) represent different goal directions in the Ant-Goal environment as defined in Figure 1(a). These were included in Figure 1(e) to illustrate that our method accurately reflects structured goal alignment. To improve comparison, we will add the same labeling to Figures 1(b–d) and enhance the caption to clarify this design.
>
> - "allows extrapolation beyond the original latents": This refers to interpolating task latents using coefficients $\alpha$ beyond the $[0,1]$ range, effectively generating virtual tasks that lie outside the convex hull of the training task distribution. This helps the agent learn from extrapolated (OOD) regions.
>
> - "[...] virtual contexts generalize to unseen tasks": This sentence explains the rationale for generating virtual contexts. Since real samples from unseen tasks are unavailable, the generated contexts do not correspond to any specific real task. As the reviewer correctly inferred, this is closer to the latter interpretation, virtual contexts are generated by decoding linearly interpolated task latents using the decoder’s generalization capability. We will clarify this point in the revised version.
>
> **3. Other Comments and Typos:**
>
> We are grateful for the reviewer’s careful reading. The following minor issues will be addressed in the revised manuscript:
>
> - Expand "LDM" to "Latent Dynamics Mixture" at first mention.
> - Correct phrasing such as “identify several key issues” to "identify two key issues."
> - Fix typos like "aallows" and remove unnecessary commas.
> - Introduce the term "task decoder" earlier in Section 2.2 for clarity.
>
> Once again, we sincerely thank the reviewer for their constructive feedback and positive recommendation. We believe these clarifications will further improve the clarity and completeness of our work.

---

> > ### Comment · Reviewer_vcob · 2025-04-09
> >
> > Thank you for your clarifications! I think that these answers reinforce my original score.

---

> > > ### Author Response · Authors · 2025-04-09
> > >
> > > Dear Reviewer,
> > >
> > > We sincerely appreciate your thoughtful and encouraging feedback. Your positive assessment means a great deal to us, and we’re truly grateful for the time and care you dedicated to reviewing our work. Thank you again for your support.

---

### Decision · Program_Chairs · 2025-05-01

**Decision:**

Accept (poster)

**Comment:**

This paper introduces Task-Aware Virtual Training (TAVT), a meta-reinforcement learning approach designed to improve generalization to out-of-distribution (OOD) tasks through a combination of metric-based task representation, virtual task generation, and sample regularization. The proposed method is positioned as an extension of prior latent-context-based methods such as PEARL, with additional components aimed at improving robustness and representation quality.

Reviewers appreciated the clarity of the paper, the extensive empirical evaluation, and the ablations that demonstrate the value of each proposed component. While some viewed the overall methodological contribution as incremental, the paper was nonetheless seen as a solid step forward in the context of OOD generalization in meta-RL.

The rebuttal effectively addressed key concerns, particularly around baseline comparisons and latent representation structure, resulting in multiple score increases post-rebuttal. Remaining reservations mostly concern the computational cost and scalability of the method, as well as its current reliance on low-dimensional task distributions.

Overall, the paper makes a meaningful contribution to the field and is suitable for inclusion at ICML.